# TRPV4 activation by TGFβ2 enhances cellular contractility and drives ocular hypertension

**Christopher Nass Rudzitis[1,2], Monika Lakk[1], Ayushi Singh[3,4], Sarah N Redmon[1], Denisa Kirdajová[1], Yun-Ting Tseng[1], Michael L De Ieso[5], W Daniel Stamer[5], Samuel Herberg[3,4,6], David Krizaj[1,2,7]***

[1]Department of Ophthalmology and Visual Sciences, Salt Lake City, United States; [2]Department of Neurobiology, University of Utah, Salt Lake City, United States; [3]Department of Ophthalmology and Visual Sciences, Syracuse, United States; [4]Department of Cell and Developmental Biology, SUNY Upstate Medical University, Syracuse, United States; [5]Department of Ophthalmology, Duke Eye Center, Duke University, Durham, United States; [6]Department of Biochemistry and Molecular Biology, SUNY Upstate Medical University, Syracuse, United States; [7]Department of Bioengineering, University of Utah, Salt Lake City, United States

**\*For correspondence:**
david.krizaj@hsc.utah.edu

## eLife Assessment

This **fundamental** work extends our understanding of the role of TGFβ2 as a modulator of mechanosensing in the eye and identifies the TRPV4 ion channel as a common regulator of Trabecular Meshwork (TM) contractility and pathological OHT and the data and evidence provided are **convincing**. This work will clearly be of interest to researchers investigating the role of mechanosensors in the TM and may underpin future research into treatments that aim to lower intra ocular pressure. This work will additionally be of interest to the growing field of researchers investigating the regulation of force sensing via ion channels and their roles in health and disease, in particular the ion channel TRPV4.

**Abstract** The risk for developing primary open-angle glaucoma (POAG) correlates with the magnitude of ocular hypertension (OHT) and the concentration of transforming growth factor-β2 (TGFβ2) in the aqueous humor. Effective treatment of POAG requires a detailed understanding of the interaction between pressure sensing mechanisms in the trabecular meshwork (TM) and biochemical risk factors. Here, we employed molecular, optical, electrophysiological, and tonometric strategies to establish the role of TGFβ2 in transcription and functional expression of mechanosensitive channel isoforms alongside studies of TM contractility in biomimetic hydrogels and intraocular pressure (IOP) regulation in a mouse model of TGFβ2-induced OHT. TGFβ2 upregulated expression of *Trpv4* and *Piezo1* transcripts and time-dependently augmented functional TRPV4 activation. TRPV4 agonists induced contractility of TM-seeded hydrogels, whereas pharmacological inhibition suppressed TGFβ2-induced hypercontractility and abrogated OHT in eyes overexpressing TGFβ2. *Trpv4*-deficient mice resisted TGFβ2-driven increases in IOP, but nocturnal OHT was not additive to TGFβ-evoked OHT. Our study establishes the fundamental role of TGFβ as a modulator of mechanosensing in nonexcitable cells, identifies the TRPV4 channel as the final common mechanism for TM contractility and circadian and pathological OHT, and offers insights for future treatments that can lower IOP in the sizeable cohort of hypertensive glaucoma patients that resist current treatments.

## Introduction

Primary open-angle glaucoma (POAG), an irreversible blinding disease, afflicts ~3.5% of the global population (*Tham et al., 2014*). Its incidence and severity are proportional to the amplitude and duration of ocular hypertension (OHT) (*Gordon et al., 2002*; *Heijl et al., 2002*), which correlates with retinal ganglion cell dysfunction, neuroinflammation, and oxidative stress (*Almasieh et al., 2012*; *Baudouin et al., 2021*). Biomechanical factors, glucocorticoids, and the cytokine transforming growth factor-β2 (TGFβ2) contribute to POAG by compromising the funneling of aqueous humor (AH) from the trabecular meshwork (TM) into Schlemm's canal (SC). Elevated intraocular pressure (IOP) enhances the contractility of the juxtacanalicular TMi, a circumocular tissue comprised of extracellular matrix (ECM) beams populated by mechanosensitive and smooth muscle-like cells—thereby increasing AH outflow resistance. The molecular mechanism that links TM pressure sensing to the contractile response is not known but is likely to underpin the tissue's sensitivity to compressive, tensile, osmotic, shear, and traction forces which collectively regulate expression of numerous TM genes and secretion of ECM proteins (*Borrás, 2003*; *Patel et al., 2021*; *de Kater et al., 1992*; *Krizaj, 2020*; *Baumann et al., 2024*; *Brubaker, 1975*; *Karimi et al., 2022*; *Patel et al., 2020*).

The increase in trabecular outflow resistance induced by mechanical stress, glucocorticoids, and TGFβ2 manifests through two distinct components: a dynamic, reversible phase amenable to cytoskeletal and Rho kinase inhibition, and a chronic phase, characterized by transdifferentiation of TM cells into fibrotic and contractile myofibroblasts (*Acott et al., 2021*; *Fuchshofer and Tamm, 2012*; *Johnstone, 2014*). TGFβ2-induced fibrotic remodeling has been linked to POAG: (1) TM cells derived from POAG patients secrete more active TGFβ2 compared to cells isolated from healthy donors (*Li et al., 2022b*), (2) the risk of POAG is proportional to [TGFβ2]$_{AH}$ (*Agarwal et al., 2015*; *Ochiai and Ochiai, 2002*; *Tripathi et al., 1994*), and (3) ectopic ocular expression of TGFβ2 suffices to induce OHT (*Patil et al., 2022*; *Shepard et al., 2010*), likely via aberrant secretion of ECM proteins and enhanced TM contractility (*Fleenor et al., 2006*; *Montecchi-Palmer et al., 2017*). The cognate TGFβ1 isoform induces similar fibrotic responses in fibroblasts, epithelial, and endothelial cells from heart, kidney, skin, and/or lung, suggesting induction of conserved fibrogenic programs (*Coker et al., 1997*; *Santiago et al., 2005*; *Yue et al., 2017*; *Zhang et al., 2020*). However, the contribution to OHT by ocular TGFβ expression cannot be disambiguated from the changing biomechanical environment: TGFβ release is activated by tissue contractility and tension (*Walker et al., 2020*; *Wipff et al., 2007*), and TGFβ activity correlates with mechanical stress gradients which may drive a cellular epithelial–mesenchymal transition-like phenotype (EMT; *Zhen et al., 2021*; *Cabrera-Benítez et al., 2012*; *Wei and Yang, 2016*).

Despite its clinical relevance, our understanding of TM mechanotransduction and its contribution to IOP homeostasis remains rudimentary. Strain and shear stress have been hypothesized to engage primary cilia and integrins, as well as mechanosensitive TRPV4, Piezo1, and TREK-1 channels (*Luo et al., 2014*; *Ryskamp et al., 2016*; *Yang et al., 2022*; *Yarishkin et al., 2021*), yet it remains unclear whether these mechanosensors regulate TM contractility, are influenced by POAG inducers like TGFβ2 or glucocorticoids, or contribute to chronic fibrosis. Among these, TRPV4 (Transient Receptor Potential Vanilloid isoform 4), a tetrameric channel with $P_{Ca}/P_{Na}$ ~ 10 (*White et al., 2016*, *Redmon et al., 2018*), is strongly expressed in rodent and human TM (*Luo et al., 2014*; *Lapajne et al., 2022*) where it carries the principal component of the pressure-activated transmembrane current and is activated by stretch, shear, and swelling (*Patel et al., 2021*; *Baumann et al., 2024*; *Ryskamp et al., 2016*; *Yarishkin et al., 2021*; *Lakk and Križaj, 2021*; *Katari et al., 2021*). Pharmacological inhibition of the TRPV4 channel and deletion of the *Trpv4* gene alter pressure gradients in the brain, kidney, lung, and bladder (*Daneva et al., 2021*; *Pochynyuk et al., 2013*; *Roberts et al., 2020*; *Redmon et al., 2018*; *Toft-Bertelsen et al., 2022*). TRPV4 mutations underpin sensorimotor neuropathies, skeletal dysplasias, retinal degeneration, and ocular dysfunction (*Nilius and Voets, 2013*; *Thibodeau et al., 2017*; *Klein et al., 2011*), while the role of TRPV4 signaling in OHT remains contentious, with evidence suggesting both IOP-lowering and IOP-elevating effects. TRPV4 dependence of conventional outflow has been linked to diverse downstream effector mechanisms (e.g., endothelial nitric oxide synthase [eNOS] and RhoA activation, phospholipid–cholesterol–caveolin regulation, OCRL inositol-5-phosphatase interaction, modulation of cell–ECM contacts, polyunsaturated fatty acid (PUFA) release, and Piezo1 signaling; *Patel et al., 2021*; *Lakk and Križaj, 2021*; *Redmon et al., 2018*; *Lakk et al., 2021*; *Uchida et al., 2021*; *Jing et al., 2024*; *Ryskamp et al., 2011*) and thus leads toward testable hypotheses: if

TM-intrinsic TRPV4 sustains steady-state normotension, promotes outflow via eNOS-dependent TM relaxation, and mitigates TGFβ2-driven fibrosis (*Patel et al., 2021*; *Lakk et al., 2021*), TRPV4 inhibition should induce OHT. Conversely, if TRPV4 activity exacerbates OHT, its blockade and deletion should reduce IOP.

In this study, we tested these hypotheses through investigation of reciprocal TRPV4–TGFβ2 interactions that perpetuate the vicious feedback loop between mechanical stressors, TM contractility, and OHT. We demonstrate that inhibition and deletion of TRPV4 lower IOP in TGFβ2 overexpression-induced and circadian OHT models and suppress TM contractility in TGFβ2-treated biomimetic hydrogels. The cytokine promoted upregulation of EMT-associated genes alongside increased transcription and activity of TRPV4, potentially sensitizing TM cells to physiological mechanical cues. Although TRPV4 activity was required to maintain OHT under physiological (nocturnal) and pathological (cytokine-induced) conditions, their respective IOP elevations were not additive, suggesting convergence on a shared common pathway that converges at the TRPV4 effector. Collectively, these findings position TRPV4 as a critical nexus of TGFβ2-induced TM contractility and IOP dysregulation. As such, TRPV4 perpetuates the vicious feedback loop between mechanical stressors and TM contractility and thus represents an ideal therapeutic target in glaucoma cases that resist current treatments.

## Results

### TGFβ2 drives overexpression of genes that encode fibrotic markers and mechanosensitive ion channels

Human TM cells respond to TGFβ2 with increased biosynthesis, deposition and degradation of ECM, altered autophagy, upregulation of F-actin stress fibers, α-smooth muscle actin (*Li et al., 2022b*; *Fleenor et al., 2006*; *Montecchi-Palmer et al., 2017*; *Nettesheim et al., 2019*; *Li et al., 2022a*), but it is unclear whether cells undergoing TGFβ2-induced fibrotic remodeling also exhibit altered capacity for sensing and transduction of mechanical stimuli. We thus profiled genes that encode known TM mechanochannels together with a selection of key cytoskeletal, ECM, and fibrotic markers in primary TM cells (pTM) isolated from three to seven donors without history of visual dysfunction (*Figure 1A–C*). Five-day exposure of pTM cells to a physiological concentration of TGFβ2 (1 ng/ml) increased the expression of EMT-promoting transcription factor SNAI1 (*SNAIL1*, p = 0.0094) and fibronectin (*FN1*, p = 0.0263), while expression of connective tissue growth factor *2* (*CCN2*, alternatively *CTGF*) was elevated in 5/5 pTM cell strains without reaching significance (p = 0.0909). Expression of fibroblast-specific protein 1 (*FSP1*, a calcium-binding fibroblast marker), yes-associated protein 1 (*YAP1*, a stiffness-induced hippo-pathway transcription factor), and *ACTA2* (αSMA, associated with cell contractility) was not consistently impacted by TGFβ2, while transcription of myocilin (*MYOC*) decreased across 4/4 pTM strains (p = 0.0055) (*Figure 1B*). Indicative of feedback inhibition (*Yan et al., 2018*), TGFβ2 treatment downregulated transcript levels of transforming growth factor beta receptor 2 (*TGFBR2*, p = 0.0219) and upregulated the expression of autoinhibitory SMAD family protein 7 (*SMAD7*, p = 0.0461) without affecting *SMAD2* or *SMAD3* expression. TGFβ2 thus promotes selective upregulation of ECM and fibrosis-related genes together with cell dedifferentiation and activation of autoregulatory SMAD mechanisms.

Analysis of genes encoding mechanosensitive channels implicated in outflow modulation (*Luo et al., 2014*; *Yarishkin et al., 2021*; *Carreon et al., 2017*; *Yarishkin et al., 2018a*) showed a 102.5% increase in expression of *TRPV4* (p = 0.0193) and a 78.9% increase in *PIEZO1* expression (p = 0.0114) across eight replicates that included seven distinct pTM strains (*Figure 1C*). Conversely, TGFβ2 exposure did not affect expression of the *TRPC1* gene (p = 0.261) and had variable, strain-dependent effects on transcript levels *of KCNK2* (p = 0.293, encoding the TREK-1 channel). Thus, TGFβ2 promotes selective transcriptional upregulation of genes that encode a subset of mechanosensitive proteins alongside fibrotic upregulation and cell dedifferentiation. Finally, we tested whether TGFβ2-induced upregulation of TRPV4 and Piezo1 is TRPV4 dependent. Inclusion of the selective TRPV4 inhibitor HC067-47 (HC-06; 5 mM), however, had no effect on transcriptional upregulation compared to TGFβ2 treatment alone (*Figure 1—figure supplement 1*).

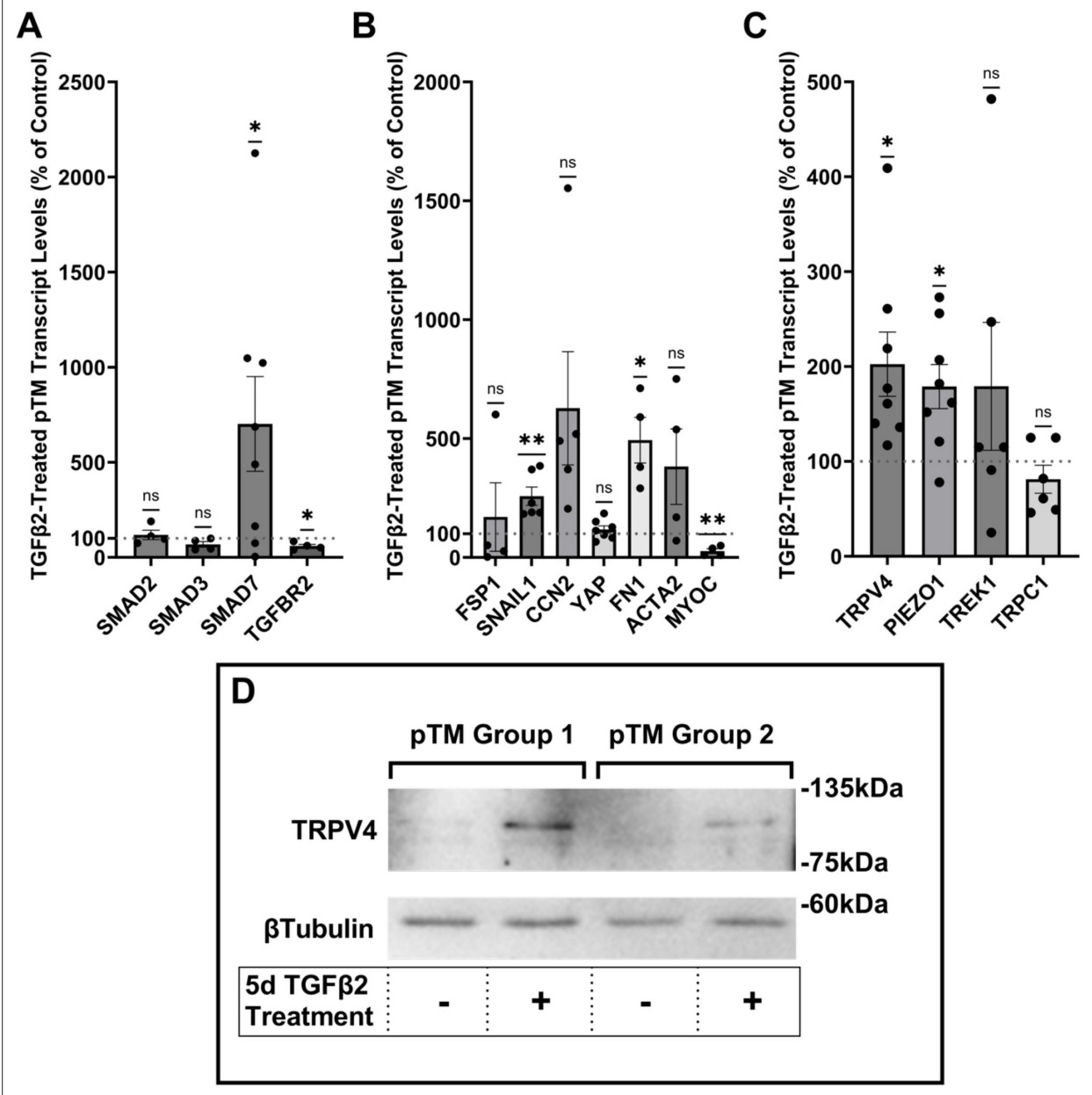

**Figure 1.** TGFβ2 induces a fibrotic phenotype in primary trabecular meshwork (pTM) cells and increases expression and membrane insertion of the TRPV4 channel. (**A, B**) Five-day TGFβ2 treatment (1 ng/ml) significantly altered expression of TGFβ pathway effectors, cytoskeletal machinery, and canonical fibrotic markers. (**C**) TGFβ2 treatment significantly increased *TRPV4* and *PIEZO1* expression, but not *TREK1* and *TRPC1* expression. Mean ± SEM shown. *N* = 4–8 experiments, each gene tested in 3–7 different pTM strains (see *Table 1*). Two-tailed one-sample *t*-test of TGFβ2-induced gene expression levels as a percent of control samples. (**D**) Isolation of membrane proteins from two separate pooled pTM samples suggests TGFβ2 treatment drives increased TRPV4 membrane insertion. *N* = 2 independent pooled samples, 3 pTM strains were pooled per sample. *p < 0.05, **p < 0.01.

The online version of this article includes the following source data and figure supplement(s) for figure 1:

**Source data 1.** Uncropped images of the membrane and HRP-signal for the western blots shown in *Figure 1* (labelled).

**Source data 2.** Uncropped images of the membrane and HRP-signal for the western blots shown in *Figure 1*.

**Figure supplement 1.** No significant difference was seen in (A) *TRPV4* or (B) *PIEZO1* expression between primary trabecular meshwork (pTM) samples treated with TGFβ2 (1 ng/ml) alone or TGFβ2 + TRPV4 antagonist HC-06 (5 µM) for 5 days.

## TGFβ2 exposure time-dependently augments TRPV4-mediated current and Δ[Ca²⁺]ᵢ

To assess the functional relevance of TGFβ2-dependent transcriptional upregulation, we determined the membrane expression and functional activation of TRPV4, which mediates the pressure-activated current and calcium signaling, regulates cytoskeletal dynamics, and modulates conventional outflow resistance in vitro (*Ryskamp et al., 2016*; *Lakk and Križaj, 2021*). TGFβ2 exposure produces an increase in levels of membrane-bound TRPV4 protein (*Figure 1D*) in western blot of two grouped pTM membrane protein samples. While low amounts of TRPV4 were visible in the membrane fractions in control samples, TGFβ2 treatment produced an increase in the higher weight TRPV4 band, suggesting

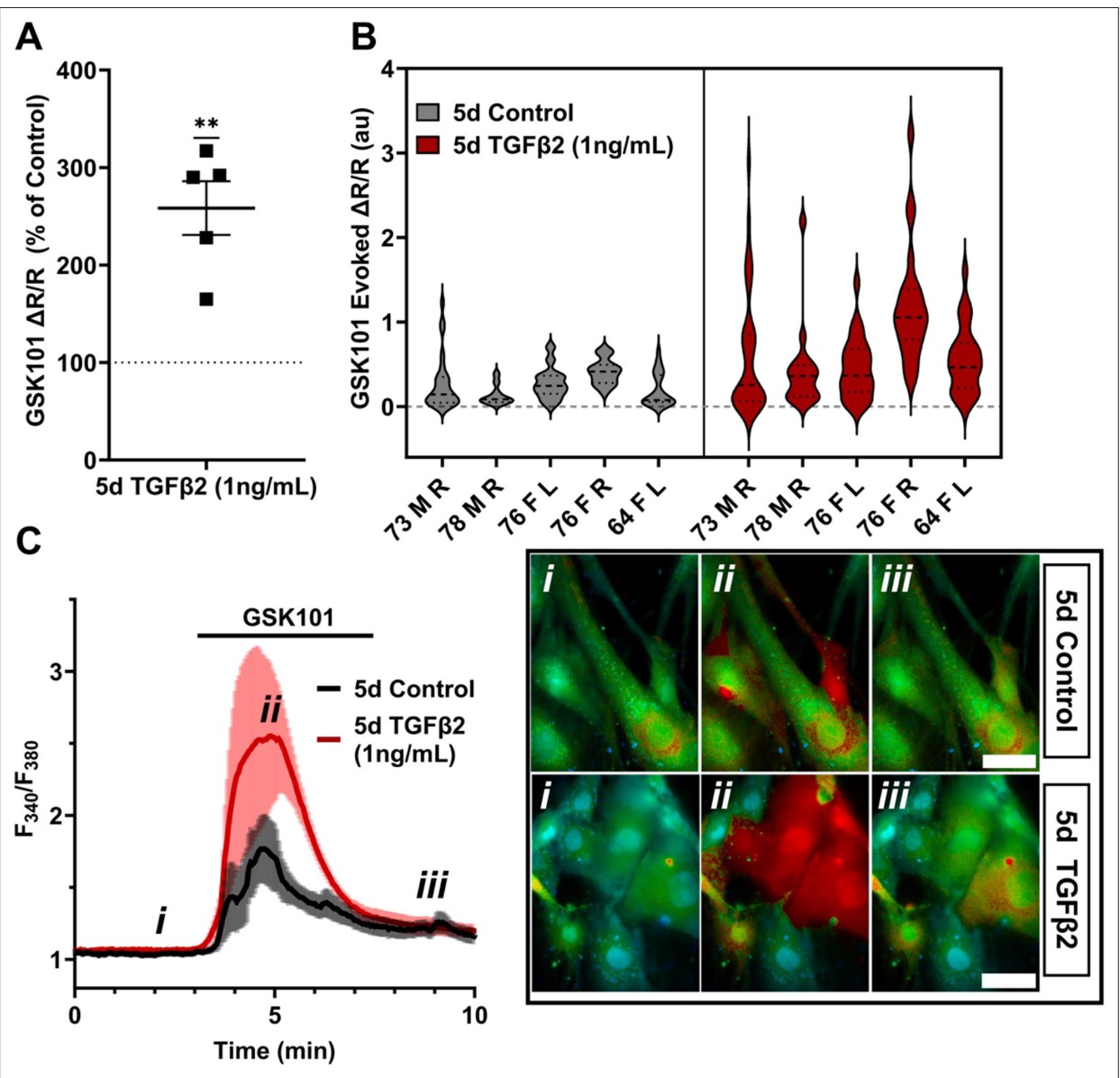

**Figure 2.** TRPV4-mediated Ca²⁺ influx is potentiated by 5-day TGFβ2 treatment. (**A**) Five-day TGFβ2 treatment (1 ng/ml) increased TRPV4 agonist-induced (GSK101, 10 nM) Ca²⁺ influx in primary trabecular meshwork (pTM) cells compared to serum-free media alone treated cells tested on the same day ($N$ = 5 pTM strains, $n$ = 3–5 slides/condition/day, individual data points over mean ± SEM). Two-tailed one-sample $t$-test of TGFβ2-treated cell average GSK101 response as a percent of control samples from the same pTM strain on the same day. (**B**) Violin plots showing the distribution of GSK101-induced Ca²⁺ responses for each pTM strain tested in A. Thick dashed line indicates mean, while light dashed line indicates quartiles. (**C**) Representative traces showing TRPV4 agonist-induced Ca²⁺ influx (seen as an increase in $F_{340}/F_{380}$) in pTM (mean ± SEM of 4 representative cells/group), alongside example Fura-2-loaded pTM cells before (**i**), during (**ii**), and after (**iii**) GSK101 application. Scale bar = 50 μm. **$p < 0.01$.

there may be isoform-specific TGFβ2-induced responses and increased TRPV4 translation leading to elevated TRPV4 trafficking, membrane insertion, and/or lipid raft interaction (*Lakk et al., 2021*).

Functional expression was assessed by tracking $[Ca^{2+}]_i$ changes in cells exposed to the selective TRPV4 agonist GSK1016790A (GSK101, 10 nM) using ratiometric Fura2-AM $Ca^{2+}$ dye, with TGFβ2-treated and control cells tested on the same day. All pTM strains responded to GSK101 with robust $[Ca^{2+}]_i$ increases which reached peak within 5 min and was followed by a gradual decrease to a steady plateau (*Figure 2C*). TGFβ2-treated cells exhibited a remarkable potentiation of GSK101-evoked $[Ca^{2+}]_i$ responses compared to control cells, with 5/5 cell strains showing an increase in the Δpeak/ baseline $F_{340}/F_{380}$ response equivalent to 258.4% ± 61.7% of the control response in (p = 0.0046) (*Figure 2A, B*). The fraction of GSK101 responders and the overall time course of responses between groups were not significantly different, indicating that TRPV4 potentiation primarily affects TRPV4-expressing cells. Thus, TGFβ2 treatment promotes TRPV4 expression and functional activity, presumably to increase their sensitivity to mechanical stressors (*Yarishkin et al., 2021*).

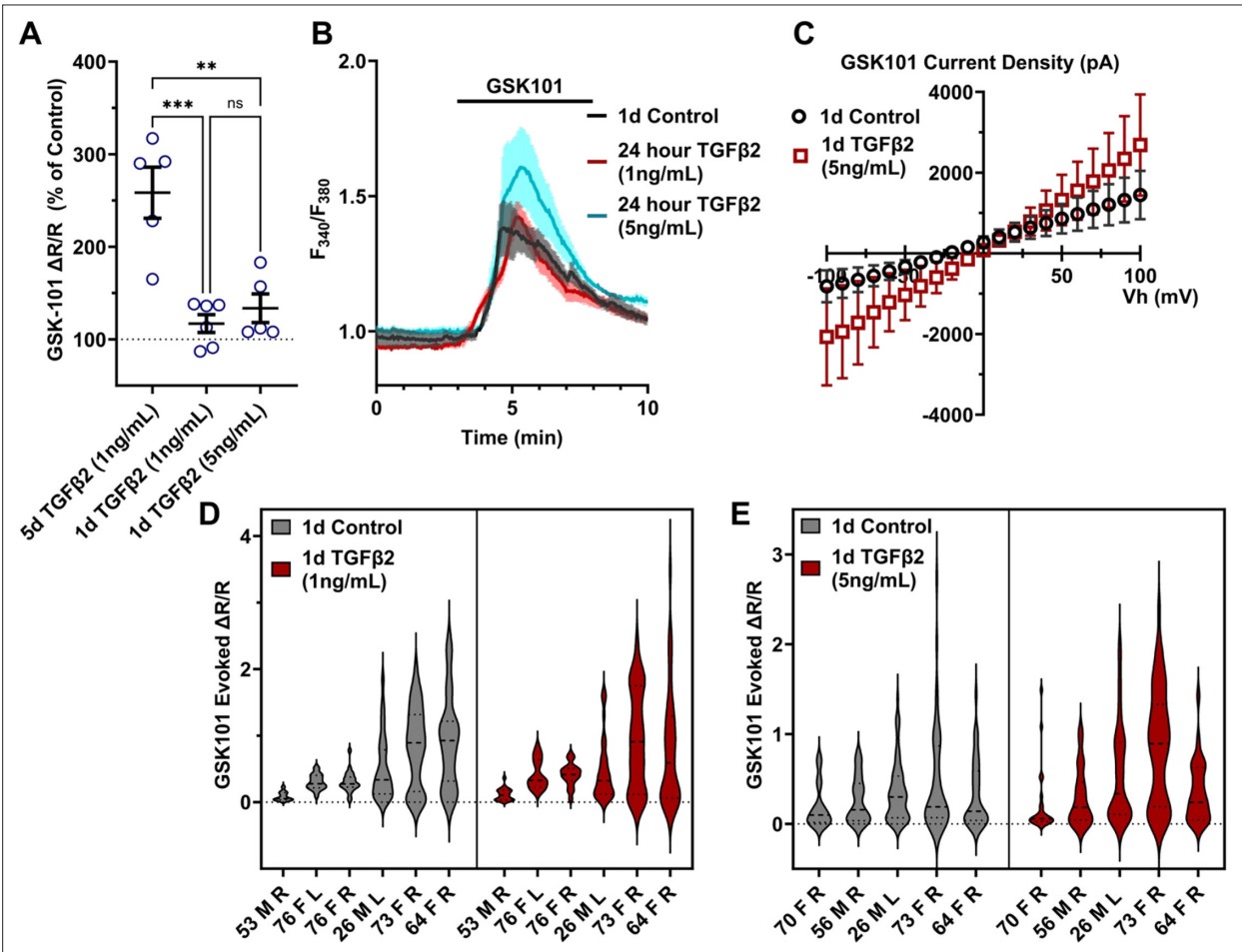

**Figure 3.** TGFβ2-induced TRPV4 potentiation is not seen at a shorter period, regardless of treatment strength. (**A**) TGFβ2 treatments for 24 hr at 1 ng/ml (N = 6 pTM strains, n = 3–5 slides/condition/day) or 5 ng/ml (N = 5 pTM strains, n = 3–5 slides/condition/day) did not show potentiation of GSK101-evoked TRPV4 $Ca^{2+}$ influx (*Figure 3—figure supplement 1*) and were significantly lower than cells treated with TGFβ2 for 5 days at 1 ng/ml (5 days TGFβ2 results from *Figure 2A*). Individual data points over mean ± SEM. One-way ANOVA with Tukey's multiple comparisons test, statistics for individual 1 day treatment groups compared to control groups shown in *Figure 1*. (**B**) Representative traces for GSK101 response following 24 hr TGFβ2 treatment, traces show mean ± SEM of 3–4 cells. (**C**) Average current density in response to GSK101 (24 hr control: n = 11 cells, 24 hr TGFβ2: n = 10 cells) shows generally increased current in TGFβ2-treated cells. Data show mean ± SEM (**D, E**). Violin plots of individual cell strains shown in **A**. Thick dashed line indicates mean, while light dashed line indicates quartiles. **p < 0.01, ***p < 0.001.

The online version of this article includes the following figure supplement(s) for figure 3:

**Figure supplement 1.** TGFβ2 concentrations of 1 ng/ml (**A**) and 5 ng/ml (**B**) did not significantly increase TRPV4-induced calcium influx with respect to control cells.

To gain insight into the time- and dose-dependence of TGFβ2-dependent TRPV4 signaling modulation, pTM cells were treated for 24 hr, at 1 and 5 ng/ml concentrations of TGFβ2. GSK101-stimulated Ca$^{2+}$ influx was not significantly increased by 24 hr TGFβ2 treatment at 1 ng/ml (Δpeak/baseline $F_{340}/F_{380}$ = 117.0% ± 23.6% of control) or 5 ng/ml (Δpeak/baseline $F_{340}/F_{380}$ = 133.6% ± 34.5% of control) *Figure 3*; *Figure 3—figure supplement 1*; the potentiation of both was significantly lower relative to the 1-day 1 ng/ml TGFβ2 treatment (p < 0.0011; *Figure 3A*). GSK101 evoked a moderately outwardly rectifying nonselective current ($I_{GSK} - I_{baseline}$) with reversal potential at ~0 mV (*Figure 3C*). While its amplitude was variable, mean current density consistently increased in cells treated for 1 day with TGFβ2 (n = 10; 5 ng/ml) relative to the control group (n = 11). The potentiating effect of TGFβ2 on TRPV4 activity appears to be time dependent, reaching significance after chronic exposure to relatively low-dose TGFβ2.

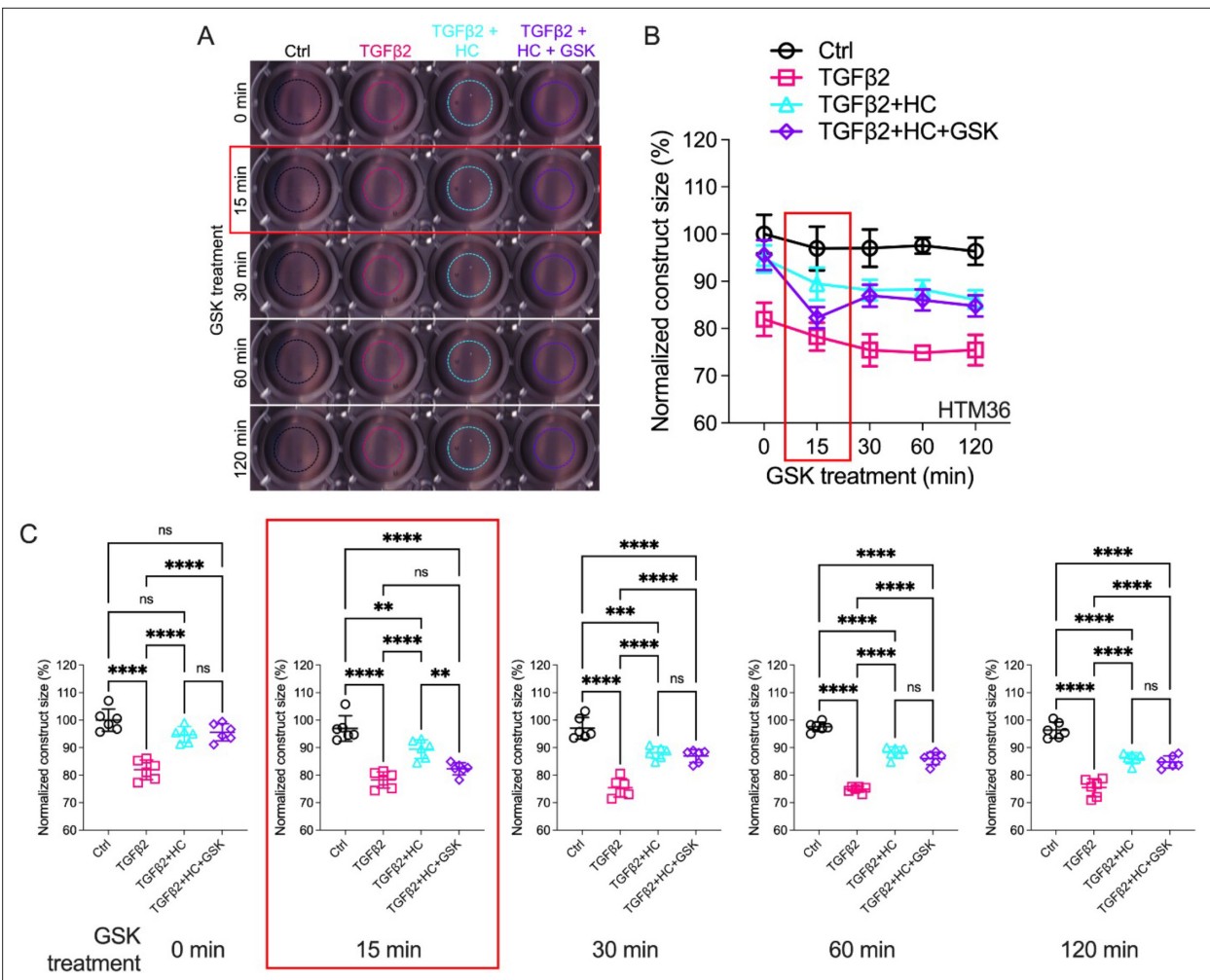

**Figure 4.** Effects of TRPV4 inhibition/activation on TGFβ2-induced contraction of trabecular meshwork (TM) cells. (**A**) Representative longitudinal 24-well plate scans of collagen type I hydrogels seeded with primary TM (pTM) subjected to the different treatments (dashed lines outline size of contracted constructs). (**B**) Longitudinal quantification of hydrogel construct size compared to the control group at the 0 min time point. (**C**) Detailed comparisons between groups at each experimental time point. n = 6 hydrogels/group. One-way ANOVA with Tukey's multiple comparisons test, data in (**B, C**) show individual data points over mean ± SD. One pTM strain shown: TGFβ2-induced contractility induction, HC-06-mediated rescue of hypercontractility, and GSK101-induced transient (15 min) contraction were consistent across (3/3) pTM strains tested (*Figure 4—figure supplement 1*). **p < 0.01, ***p < 0.001, ****p < 0.0001.

The online version of this article includes the following figure supplement(s) for figure 4:

**Figure supplement 1.** TRPV4 activation is obligatory for TGFβ2-induced trabecular meshwork (TM) cell contractions.

**Figure supplement 2.** Detailed view of primary trabecular meshwork (pTM) seeded collagen constructs used in *Figure 4* and *Figure 4—figure supplement 1*.

## TGFβ2-induced TM contractility requires TRPV4 activation

The IOP-lowering effectiveness of Rho kinase inhibitors and latrunculins (*Li et al., 2022a*; *Honjo, 2001*; *Rao et al., 2017*; *Ethier et al., 2006*) indicates that sustained increases in outflow resistance require tonic actin polymerization and contractility. TGFβ2 drives the TM myofibroblast contractile response (*Li et al., 2022a*), while the role of mechanosensation remains unknown. To ascertain whether TRPV4 upregulation (*Figures 1 and 2*) contributes to the contractile response, we seeded pTM cells into high-compliance type I collagen hydrogels (*Li et al., 2022a*; *Figure 4—figure supplements 1 and 2*). Hydrogels that were incubated with TGFβ2 (5 ng/ml) exhibited marked increases (p < 0.0003) in the rate and the magnitude of contraction at all time points (*Figure 4*, *Figure 4—figure supplements 1 and 2*). Simultaneous treatment with HC-06 (5 μM) significantly reduced the extent of TGFβ2-induced TM contractility (p < 0.0001). To determine whether TRPV4 activation is sufficient to induce the contractile response, the antagonist was washed out and hydrogels supplemented with GSK101 (25 nM). Fifteen minutes post-treatment, the constructs responded to the agonist with transient contraction (*Figure 4C*; *Figure 4—figure supplement 1*, p < 0.01), with a time course mirroring GSK101-induced $[Ca^{2+}]_i$ elevations (*Figures 2 and 3*). The effects of TRPV4 inhibition and activation were consistent across all pTM strains tested (N = 3). TRPV4-mediated $Ca^{2+}$ influx is therefore sufficient to induce TM contractility and necessary for pTM hypercontractility induced by TGFβ2.

## TRPV4 activity is required to maintain TGFβ2-induced OHT

To test whether TRPV4 contributes to TGFβ2-induced OHT in vivo, we utilized the lentiviral TGFβ2 overexpression model developed by *Patil et al., 2022* in which adult C57BL/6J mice (N = 5) were intravitreally injected with lentivirus overexpressing constitutively active human TGFβ2 (LV-TGFβ2). LV-TGFβ2-injected eyes, but not the contralateral eyes injected with a lentivirus containing a scrambled transgene (LV-Ctrl), exhibited significant IOP elevations 1-week post-transduction (*Figure 5A*, Week 2, $\Delta_{TGF-Ctrl}$ = 4.0 mm Hg, p = 0.0143). By 2 weeks post-transfection, IOP in LV-TGFβ2 eyes reached 19.9 ± 4.7 mm Hg, whereas IOP in LV-Control eyes remained at control levels (14.0 ± 1.2 mm Hg), with $\Delta_{TGF-Ctrl}$ = 5.9 mm Hg (p = 0.0002). IOP remained elevated throughout the 4 weeks after the injection (Week 5, $\Delta_{TGF-Ctrl}$ = 4.9 mm Hg, p = 0.0008). HC-06 (100 μM) microinjection into the anterior chamber of LV-TGFβ2 and LV-Ctrl eyes lowered IOP in LV-TGFβ2 eyes to 12.2 ± 1.7 mm Hg after 24 hr ($\Delta_{postHC-preHC}$ = −5.8 mm Hg) with no difference observed in IOP from LV-Ctrl eyes (12.6 ± 1.9 mm Hg, $\Delta_{postHC-preHC}$ = −0.3 mm Hg). LV-Ctrl eyes remained close to pre-injection levels post-HC-06 treatment (*Figure 5A, B*). IOP in LV-TGFβ2 eyes returned to hypertensive levels by 1 week post-HC-06 injection (Weeks 6 and 7, $\Delta_{TGF-Ctrl}$ = 3.9 mm Hg, p = 0.0201). To determine the effect of the bolus injection alone, LV-TGFβ2 and LV-Ctrl eyes were reinjected with PBS 2 weeks after re-establishing the OHT baseline. The sham injection transiently reduced IOP in LV-TGFβ2 ($\Delta_{postPBS-prePBS}$ = −4.5 mm Hg) and LV-Ctrl ($\Delta_{postPBSpre-PBS}$ = −1.2 mm Hg) eyes; however, LV-TGFβ2 eyes returned to hypertensive levels by 48 hr post-injection ($\Delta_{TGF-Ctrl}$ = 3.6 mm Hg, p = 0.0465) and to pre-injection levels after 72 hr ($\Delta_{TGF-Ctrl}$ = 5.4 mm Hg, p = 0.0002). Bolus injection was less effective than HC-06 at all time points 24 hr post-injection (Weeks 8 and 9, *Figure 5B*). These data indicate that selective pharmacological inhibition of TRPV4 effectively and reversibly blocks TGFβ2-induced OHT.

To further evaluate the TRPV4 dependence of TGFβ-induced OHT, we took advantage of mice with global *Trpv4* knockdown (*Liedtke and Friedman, 2003*; *Ryskamp et al., 2011*; *Yarishkin et al., 2018b*). $Trpv4^{-/-}$ mice (N = 6) were injected with LV-TGFβ2 and LV-Ctrl vectors in contralateral eyes (*Figure 5C*). Additionally, two littermate control mice injected alongside the $Trpv4^{-/-}$ animals were added to previously collected WT LV-injected cohorts measured at the same timepoints (N = 8–15, *Figure 5C*). Pre-LV injection, IOP levels in $Trpv4^{-/-}$ animals were comparable to the WT cohort, indicating that TRPV4 activity does not regulate normotension. Similarly, IOP in LV-Ctrl-injected eyes was not significantly different between WT and $Trpv4^{-/-}$ animals at any point in the experiment (peak $\Delta_{CtrlKO-CtrlWT}$ = −1.2 mm Hg, *Figure 5D*, *Figure 5—figure supplement 1*). By 2 weeks post-injection (Week 3), LV-TGFβ2-treated $Trpv4^{-/-}$ eyes exhibited significantly lower IOP compared to the LV-TGFβ2 WT cohort ($\Delta_{TGFKO-TGFWT}$ = −3.1 mm Hg, p = 0.0009, *Figure 5C*). LV-TGFβ2 injected $Trpv4^{-/-}$ eyes exhibited mild OHT but the effect was significantly reduced compared to WT eyes, and IOP decreased by 2 weeks post-injection (*Figure 5C, D*).

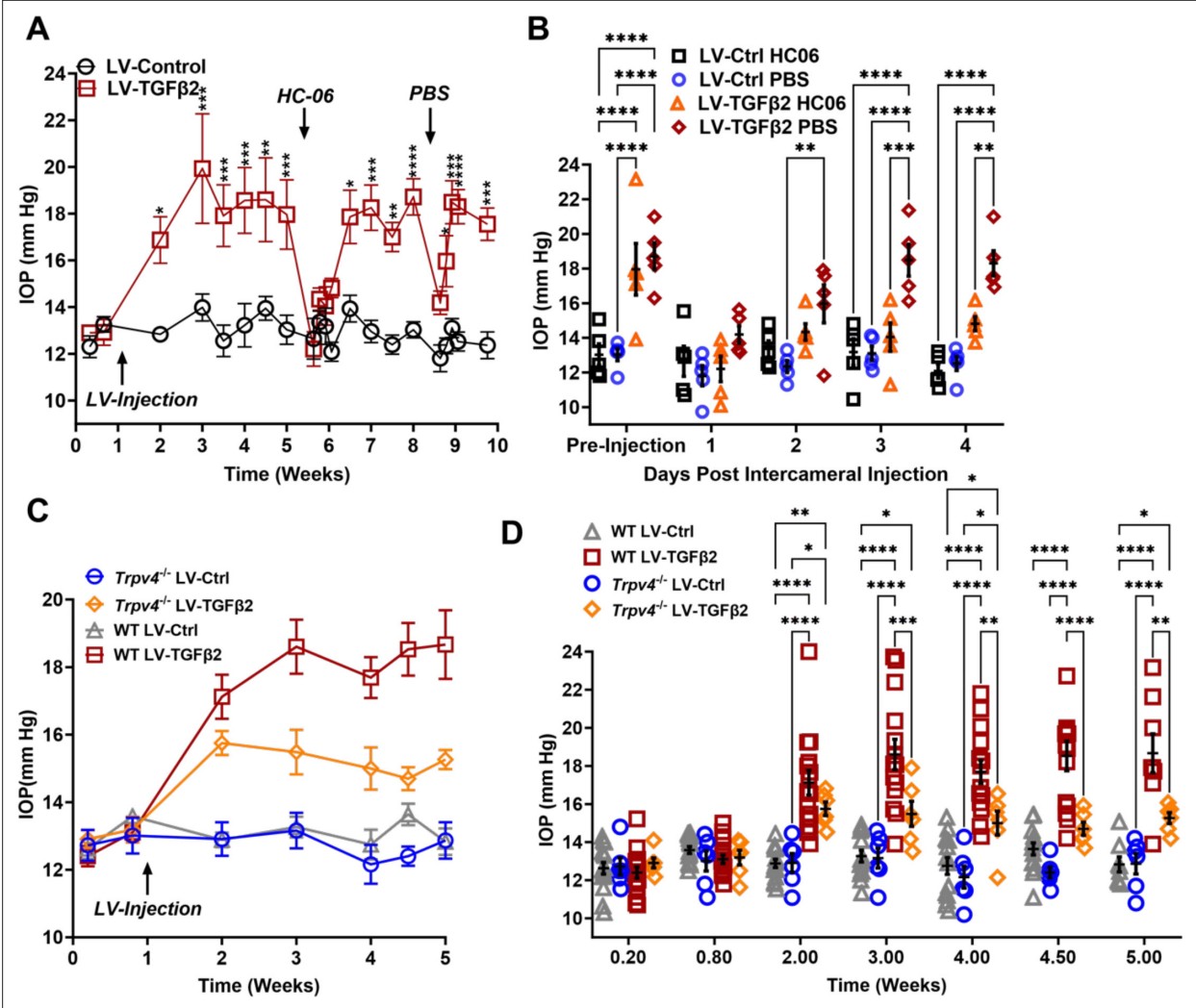

**Figure 5.** TRPV4 activation is necessary to maintain LV-TGFβ2-induced ocular hypertension (OHT). (**A**) Intravitreal injection of LV-TGFβ2 (Week 1), but not LV-Control, elevates intraocular pressure (IOP) in WT mice (*N* = 5 eyes/group) as early as 1-week post-injection. Injection of TRPV4 antagonist HC-06, but not PBS, produced multiday IOP reduction in LV-TGFβ2-treated eyes. HC-06 and PBS injections did not affect IOP in LV-Control injected eyes. Two-way ANOVA with Bonferroni post hoc analysis (**B**) Direct comparison of the results of PBS and HC-06 injections in the eyes shown in **A**. Two-way ANOVA with Bonferroni post hoc analysis. (**C**) Intravitreal injection of LV-TGFβ2 in *Trpv4⁻/⁻* mice (*N* = 6 eyes/group) resulted in only mild OHT; plotted against WT eyes at matching timepoints (3 WT cohorts including the 5 WT eyes shown in A, B, *N* = 8–15 eyes/group). (**D**) Statistical comparison of the IOP values shown in **C**. The IOP in LV-TGFβ2 WT eyes was significantly elevated compared to the LV-TGFβ2 *Trpv4⁻/⁻* eyes from 2 weeks post-injection. LV-Control injected eyes in WT or *Trpv4⁻/⁻* eyes remain close to the baseline value and are not significantly different. Two-way ANOVA with Bonferroni post hoc analysis. (**A, C**) shows mean ± SEM. Data in (**B, D**) show individual data points over mean ± SEM, *p < 0.05, **p < 0.01, ***p < 0.001, ****p < 0.0001.

The online version of this article includes the following source data and figure supplement(s) for figure 5:

**Source data 1.** Source data for Lv-Control IOP and Lv-TFFb2 cohorts treated with HC-06.

**Source data 2.** Source data for IOP data from WT and Trpv4 KO eyes treated with TGFβ2.

**Figure supplement 1.** Expansion of *Figure 5D*.

## TGFβ2-induced and nocturnal OHT are non-additive but require TRPV4

Mammalian IOP is modulated by the circadian rhythm, with levels elevated at night and nocturnal IOP fluctuations implicated in POAG (*Patel et al., 2021*; *Redmon et al., 2024*; *Ikegami and Masubuchi, 2022*). We measured nocturnal (9:00–10:00 PM) IOP in LV-TGFβ2 (*N* = 4) and LV-Ctrl WT eyes (*N* = 4) from isoflurane-anesthetized mice ~2 months post-LV injection to determine whether nocturnal OHT is additive to TGFβ2-induced elevation observed during the daytime (12:00–2:00 PM, *Figure 6A*). LV-TGFβ2 injected eyes showed significant IOP elevation compared to LV-Ctrl eyes during daytime

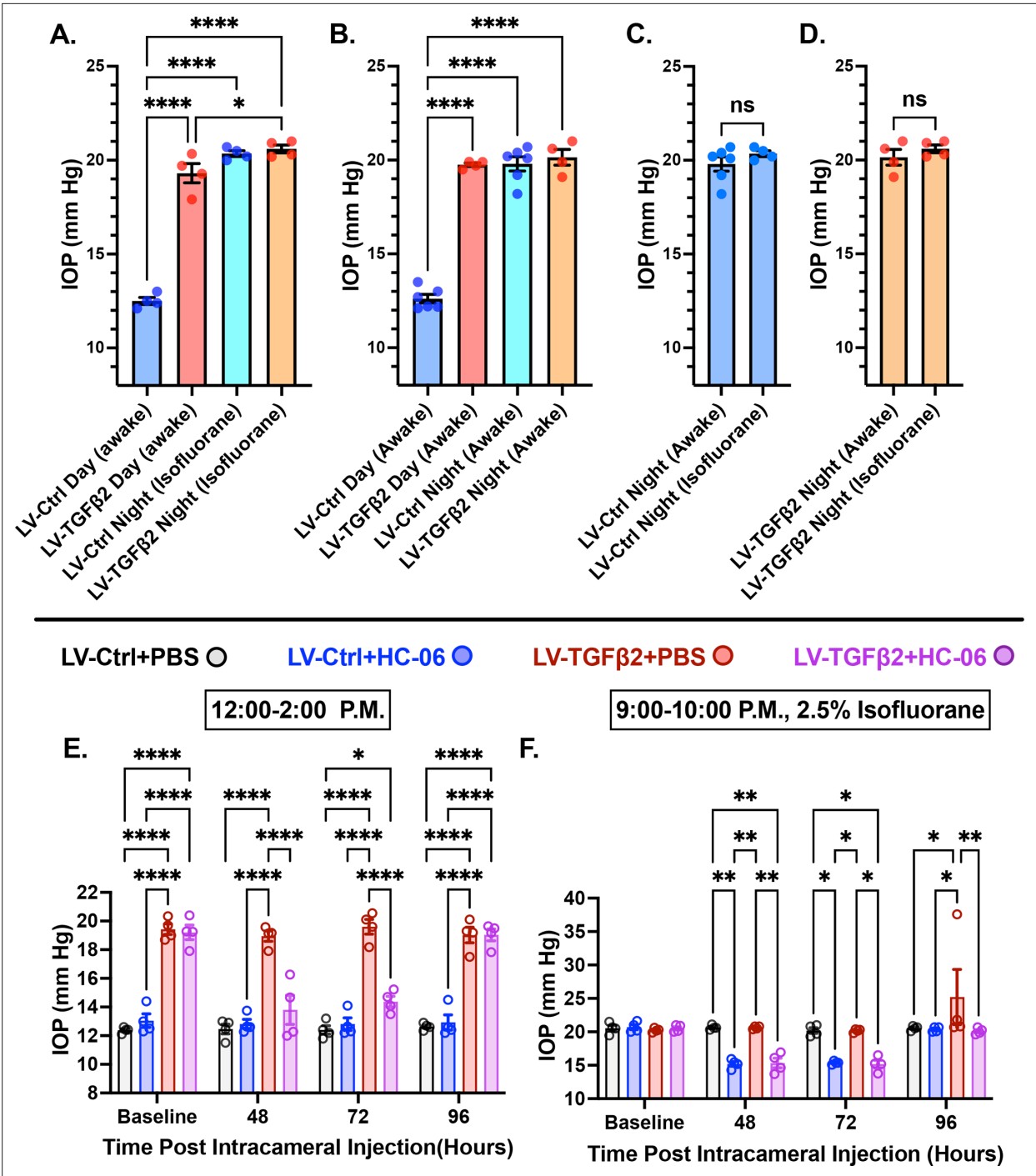

**Figure 6.** TRPV4 inhibition inhibits nocturnal intraocular pressure (IOP) elevation in control and TGFβ2 overexpressing eyes. (**A, B**) Post-LV injection daytime (12–2:00 P.M) and nocturnal (9–10:00 P.M.) IOP compared in WT mice (*N* = 4–6 eyes/group) before drug treatment. LV-TGFβ2 eyes were elevated at daytime, but nocturnal ocular hypertension (OHT) was not significantly different between LV-Ctrl and LV-TGFβ2 eyes in two separate cohorts of mice measured under isoflurane anesthesia (**A**) or while awake (**B**). (**C, D**) Anesthesia had no significant effect on measured nocturnal IOP. One-way ANOVA with Tukey's multiple comparisons test. (**E, F**) PBS-injected eyes did not exhibit changes in daytime or nighttime intraocular pressure; however, HC-06 injection reduced TGFβ2-induced IOP elevations during the day and LV-Ctrl and LV-TGFβ2 nocturnal IOPs (*N* = 4 eyes/group); two-way ANOVA with Bonferroni post hoc analysis. Figures show data points over mean ± SEM, *p < 0.05, **p < 0.01, ****p < 0.0001.

The online version of this article includes the following source data and figure supplement(s) for figure 6:

**Source data 1.** Source data for nocturnal and diurnal IOP cohorts from control mice and animals treated with LV-TGFβ2.

**Figure supplement 1.** Nocturnal intraocular pressure (IOP) is not significantly affected by LV-TGFβ2 overexpression.

measurements (diurnal $\Delta_{TGF\beta-Ctrl}$ = 7.9 mm Hg, p < 0.0001) but the difference vanished at night (nocturnal $\Delta_{TGF\beta-Ctrl}$ = 0.2 mm Hg), indicating that TGFβ2-induced OHT is not additive to the circadian OHT. To test whether the IOP measurement was influenced by isoflurane-induced anesthesia, we repeated the nocturnal measurements in awake animals (N = 4–6 eyes/group). We observed no difference in nocturnal IOP between the two groups of animals was observed (*Figure 6A–D*, *Figure 6—figure supplement 1*). To determine whether physiological (nocturnal) OHT requires TRPV4, we microinjected LV-Ctrl (N = 4) and hypertensive LV-TGFβ2 (N = 4) eyes with PBS or HC-06. PBS injection did not affect IOP in LV-Ctrl or LV-TGFβ2 eyes at day or night (*Figure 6E, F*) except for a single LV-TGFβ2 eye that exhibited abnormally high nocturnal IOP (37 mm Hg) at the 4-day timepoint. Conversely, HC-06 injection blocked LV-TGFβ2-induced IOP during the day (P<0.001) and significantly lowered IOP in LV-Ctrl and LV-TGFβ2 eyes at night (~5 mm Hg; p < 0.01). These data indicate that (1) TRPV4 activation is necessary for OHT in the TGFβ2 overexpression mouse model (*Figures 5 and 6*) and the circadian IOP elevations, (2) TGFβ2-evoked OHT does not affect nocturnal IOP elevation in mice, and (3) TRPV4 inhibition does not disrupt the mechanisms that maintain daytime normotensive IOP (*Figures 5 and 6*).

## Discussion

This study establishes a mechanistic framework that integrates biochemical and biomechanical risk factors of POAG and highlights the pivotal role of TRPV4, a polymodal $Ca^{2+}$-permeable channel, as a key regulator of TM contractility and OHT. Our central finding is that the glaucoma-inducing cytokine TGFβ2 amplifies TRPV4 expression and activity, which in turn drives tonic increases in TRPV4 activation and TM contractility that are required to maintain elevated IOP. These observations position the pressure-activated channel (*Yarishkin et al., 2021*) as a molecular linchpin that links mechanical stress to neurodegeneration resulting from the obstruction of the primary outflow pathway. Our confirmation of the linkage between TGFβ signaling and TRPV4 in TM links POAG pathophysiology to fibrotic remodeling seen in other tissues (*Rahaman et al., 2014*; *Willard et al., 2021*). Considering that current glaucoma treatments target secondary outflow mechanisms or incur side effects (such as hyperemia) (*Sharif, 2023*; *Weinreb et al., 2002*), the IOP lowering achieved through TRPV4 inhibition and gene knockdown promises a novel therapeutic avenue to mitigate ocular injury.

The etiology of glaucoma reflects convergence of risk factors that include IOP and TGFβ2: epidemiological data correlate the incidence of POAG with the amplitude of IOP and [TGFβ2]$_{AH}$ (*Tripathi et al., 1994*; *Leske et al., 2003*), while chronic increases of either [TGFβ2]$_i$ or IOP promote fibrotic remodeling of TM/SC and augment the flow resistance of the conventional pathway (*Fuchshofer and Tamm, 2012*; *Shepard et al., 2010*; *Fleenor et al., 2006*). TGFβ2-induced facility suppression has been historically attributed to changes in composition, crosslinking, and amount of ECM (*Fleenor et al., 2006*; *Montecchi-Palmer et al., 2017*; *Fuchshofer et al., 2005*; *Medina-Ortiz et al., 2013*), activation of Hippo signaling and Rho kinase- (Rho/ROCK) mediated contractility (*Li et al., 2022b*; *Li et al., 2022a*) and altered expression of genes encoding mitogen-activated protein kinase, immune response, oxidative stress, and/or ECM pathways (*Callaghan et al., 2022*; *Inoue-Mochita et al., 2015*; *Kang et al., 2013*). Our discovery reveals that such biochemical mechanisms engage in bidirectional interplay with biomechanical factors and transducers: TGFβ2 impacts the expression and function of TM mechanosensors and vice versa, TRPV4 is required for TGFβ2-induced contractility. Specifically, we found that TGFβ2 (1) induced time-dependent upregulation of TRPV4 mRNA and amplified TRPV4-mediated calcium signaling, while (2) TRPV4 was required to mediate TGFβ2-induced TM hypercontractility and maintain chronic OHT in TGFβ2-treated mouse eyes. Microinjection of the selective antagonist HC-06 accordingly reduced IOP in LV-TGFβ2-treated eyes to baseline with hypotension persisting for ~4 days and reversing to pre-injection OHT by day 7. The TRPV4 dependence of TGFβ2-induced OHT and contractility was corroborated in *Trpv4*$^{-/-}$ mice and in vitro using TGFβ2-treated 3D hydrogel constructs. In vivo, pharmacological inhibition (~100% reduction in OHT, transient) outperformed gene knockdown (~50% reduction in OHT, stable), potentially due to compensatory mechanosensory mechanisms in Trpv4-*null* animals (*Redmon et al., 2024*).

We've previously shown that TM TRPV4 is activated by physiological (5–25 mm Hg) pressure steps (*Yarishkin et al., 2021*; *Yarishkin et al., 2018a*) and (1–12%) strains (*Ryskamp et al., 2016*; *Lakk and Križaj, 2021*), which trigger downstream outflow-relevant signaling through Rho kinase, F-actin, tyrosine phosphorylation of FAK, paxillin and vinculin, lipid remodeling, and ECM release mechanisms

(*Ryskamp et al., 2016*; *Lakk and Križaj, 2021*; *Lakk et al., 2021*). The present study extends those observations by revealing the TRPV4 -dependence of TM contractility (the agonist GSK101 induced, and the antagonist HC-06 suppressed, contractility in a 3D biomimetic model) and by establishing TRPV4 as an obligatory effector of OHT under physiological (circadian rhythmicity) as well as pathological conditions. These findings resolve conflicting reports of hypotensive vs. hypertensive effects of TRPV4 modulation (*Patel et al., 2021*; *Luo et al., 2014*; *Ryskamp et al., 2016*; *Yarishkin et al., 2021*; *Lakk and Križaj, 2021*; *Lakk et al., 2021*; *Uchida et al., 2021*; *Redmon et al., 2024*). While TRPV4 activity has been suggested to lower IOP via phosphoinositide signaling in primary cilia (*Luo et al., 2014*), TM-resident eNOS activity (*Patel et al., 2021*), release of PUFAs (*Uchida et al., 2021*), and/or signaling downstream from Piezo1 mechanosensing (*Jing et al., 2024*), these mechanisms are challenged by evidence that TRPV4-regulated $Ca^{2+}$ influx persists in TM cells with ablated primary cilia (*Ryskamp et al., 2016*), eNOS expression in TM cells is vanishingly low (*Reina-Torres et al., 2021*; *van Zyl et al., 2020*), PUFAs such as arachidonic acid and epoxyeicosatrienoic acids stimulate rather than inhibit TRPV4 (*Ryskamp et al., 2016*), and TRPV4 signaling in TM cells is unaffected by Piezo1 inhibition and knockdown (*Yarishkin et al., 2021*). Rather, the suppression of outflow facility by Piezo1 inhibitors in vitro and in vivo (*Yarishkin et al., 2021*; *Zhu et al., 2021*) suggests that Piezo1 opposes the hypertensive functions of TRPV4.

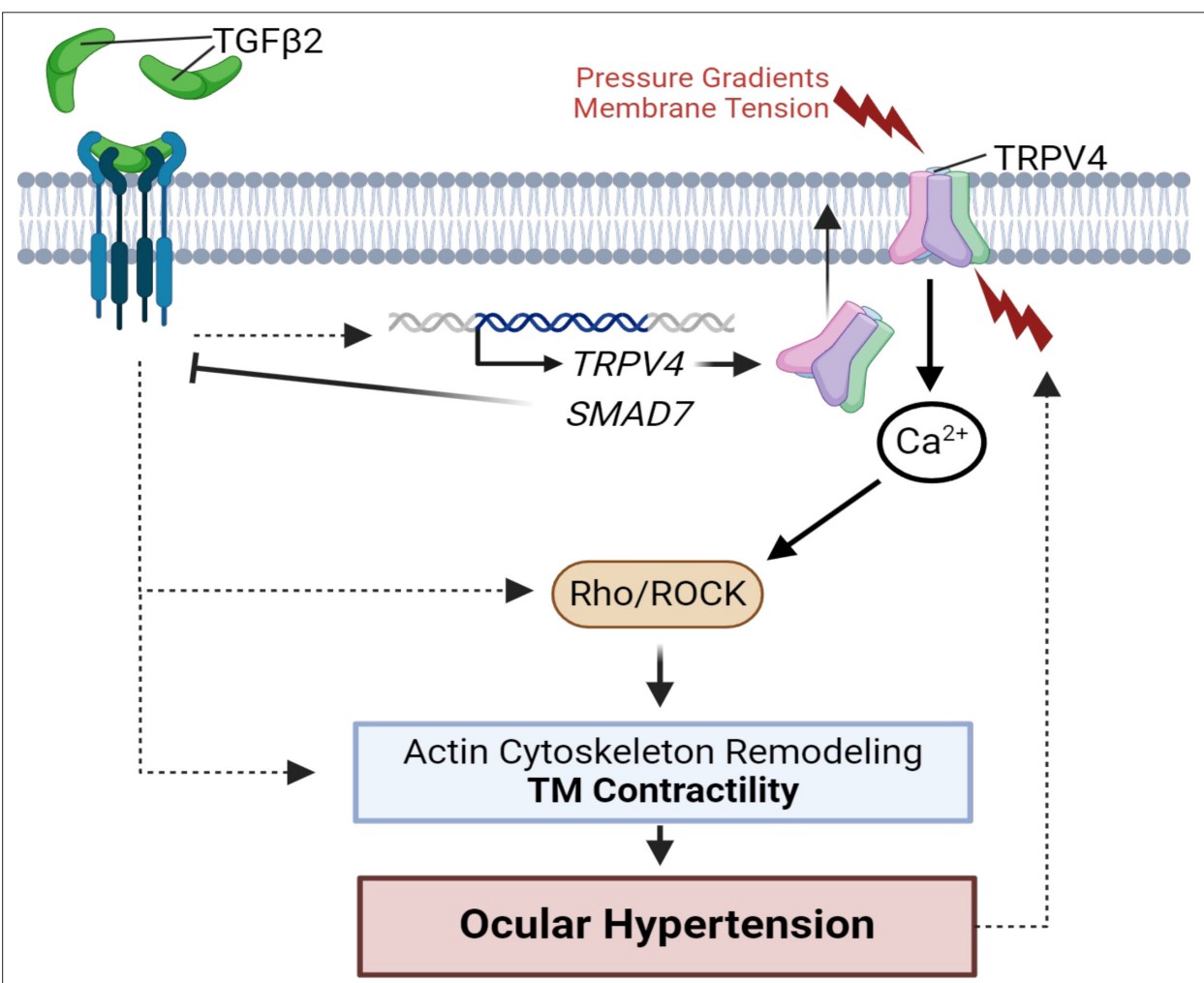

**Figure 7.** TGFβ2–TRPV4 interactions in trabecular meshwork (TM) remodeling and ocular hypertension (OHT). Chronic exposure to TGFβ2 induces upregulation of functional TRPV4 channels alongside the autoinhibitory canonical modulator SMAD7. TRPV4-mediated $Ca^{2+}$ influx, canonical, and non-canonical TGFβ2 signaling stimulate the Rho/ROCK pathway to augment cytoskeletal contractility and stimulate extracellular matrix (ECM) release. Actomyosin contractility promotes outflow resistance and drives OHT and underpins a vicious feedforward TRPV4-dependent loop that maintains OHT. This figure was created using BioRender.com.

The TRPV4 dependence of outflow resistance is indicated by multiple lines of evidence. In biomimetic human TM scaffolds that support flow devoid of ciliary body or SC influences, TRPV4 inhibition enhanced, and activation suppressed, the facility (*Ryskamp et al., 2016*). We demonstrated that the agonist (GSK101) induces contractility while the antagonist (HC-06) mitigated TGFβ2-induced hypercontractility and lowered IOP in TGFβ2-overexpressing eyes. These results support a model wherein TGFβ2 augments TRPV4-mediated pressure transduction to drive hypercontractility and fibrosis via Ca$^{2+}$- and Rho-dependent stress fiber formation and reinforcement of focal ECM contacts (*Lakk and Križaj, 2021*; *Pattabiraman et al., 2015*; *Zhang et al., 2008a*; *Figure 7*). Taking into account TRPV4 thermosensitivity (Q$_{10}$ of ~10, with peak activation at physiological temperature of ~34–37°C; *Güler et al., 2002*; ; *Nishimoto et al., 2021*), TGFβ2-induced contractility of TM-populated hydrogels may reflect the synergy between severalfold transcriptional and functional amplification of TRPV4-mediated signaling (*Figures 1–3*) and constitutive temperature-facilitated activation of TRPV4. These synergistic effects were unmasked as the absence of hypercontractility in samples treated with HC-06. The residual contractility in HC-06-treated cells could reflect contributions from Piezo1, TRPC, and/or TREK-1 channels (*Ryskamp et al., 2016*; *Yarishkin et al., 2018a*; *Abad et al., 2008*; *Feger et al., 2017*). Parallels from studies conducted on heart, lung, liver, skin, and articular cartilage similarly found that TRPV4 contributes to the progression of fibrosis induced by the cognate TGFβ1 cytokine (*Rahaman et al., 2014*; *Sharma et al., 2017*; *Adapala et al., 2013*; *O'Conor et al., 2014*; *Song et al., 2014*) and implicated the TRPV4 channel in the contractility of the bladder (*Wu et al., 2021*), heart (*Jones et al., 2019*; *Chaigne et al., 2023*), and blood vessels (*Wen et al., 2023*) . Interestingly, conditional *Trpv4* ablation from smooth muscle cells lowered blood pressure (*Chen et al., 2022*; *Zhu et al., 2023*), an effect not dissimilar from IOP lowering in *Mgp:Trpv4* cKO mice (*Figure 5*).

TGFβ2-induced upregulation of *FN1*, *SNAIL1*, and *CTGF* transcripts (*Figure 1*) accords with RNA profiling studies which cataloged the cytokine's role in transdifferentiation of TM cells into contractile myofibroblasts (*Fleenor et al., 2006*; *Callaghan et al., 2022*; *Zhao et al., 2004*; *Zhavoronkov et al., 2016*; *Last et al., 2011*; *Raychaudhuri et al., 2017*; *Read et al., 2007*; *Wang et al., 2017*) whereas the decreased expression of the *TGFBR2* gene and increased abundance of *SMAD7* mRNA indicate concurrent activation of autoinhibitory mechanisms associated with canonical TGFβ-family signaling (*Fuchshofer et al., 2009*). The threefold increase in *TRPV4* transcription and responsiveness to GSK101 was observed at POAG-relevant TGFβ2 concentrations (in AH, ~0.2–3.2 ng/ml; *Agarwal et al., 2015*), with a time course that mirrored facility reduction in human eyes treated with exogenous cytokine (*Gottanka et al., 2004*). A single 5 ng/ml TGFβ2 dose was sufficient to double the amplitude of the GSK101-evoked current and alter its rectification (*Figure 3*), with imaging experiments confirming robust and reproducible increases in Ca$^{2+}$ signals across the 5/5 studied strains after 5 days of treatment (*Figure 2*). The effects of TGFβ2 on $I_{TRPV4}$, membrane protein levels and [Ca$^{2+}$]$_{GSK}$ accord with increased expression of the *TRPV4* gene. Precedents from other cell types (e.g., fibroblasts) suggest functional upregulation might involve increased trafficking of TRPV4–PI3Kγ complexes and/or β-arrestin 1-dependent ubiquitination (*Grove et al., 2019*; *Shukla et al., 2010*). The upregulation of *TRPV4/PIEZO1* transcription by TGFβ2 predicts exaggerated responsiveness of the outflow pathway to mechanical loading, as reported for TRPV4-dependent mechanical hyperalgesia in chemotherapy (*Matsumura et al., 2014*), neuropathic pain (*Zhang et al., 2008b*; *Alessandri-Haber et al., 2006*), cancer (*Maqboul and Elsadek, 2018*), and diabetic neuropathy (*Cui et al., 2020*).

An important corollary of these findings is that they extend TRPV4's role beyond pathology into the IOP homeostasis in healthy animals. TRPV4 inhibitors, ROCK inhibitors, and TM-specific expression of dominant negative scAAV2.*dnRhoA* constructs lower IOP across diverse OHT models (occlusion of the iridocorneal angle, TGFβ, glucocorticoids, and the nocturnal cycle; *Redmon et al., 2024*; *Rao et al., 2017*; *Borrás et al., 2015*). The TRPV4 dependence and lack of additivity between physiological (circadian-induced) and pathological OHT modes (*Figure 6*) in untreated and TGFβ2 eyes imply convergence at the level of TRPV4-Rho signaling as the final effector of conventional outflow resistance. An interesting challenge for future studies will be elucidation of mechanisms that underlie the reversibility of circadian TRPV4 activation, which might combine testing involvement of the suprachiasmatic nucleus, the hypothalamus–pituitary–adrenal axis (*Samuels et al., 2012*; *Ikegami, 2024*), nocturnal release of norepinephrine and melatonin (*Shen et al., 1992*; *Steinberg et al., 2015*) and circadian TRPV4 modulation via β1 integrins (*Matthews et al., 2010*), caveolin-1 (*Lakk et al., 2021*), and cytoskeletal proteins (actin, actin adaptor proteins, microtubules) (*Goswami et al., 2010*).

In conclusion, this study bridges biomechanical and biochemical paradigms of glaucomatous remodeling by extending the role of TGFβ2 beyond fibrosis to include TRPV4-dependent signaling and actomyosin contraction. We propose that TGFβ2-induced upregulation of TRPV4 expression shifts the normotensive setpoint maintained by steady-state TRPV4, Piezo1, and TREK-1 activity (*Yarishkin et al., 2021*; *Yarishkin et al., 2018a*; *Zhu et al., 2021*) to heighten the sensitivity of the conventional outflow pathway to mechanical cues. Mechanistically, this involves the hijacking of the contractile machinery and increased fibrotic remodeling, which in turn promote the pull of stress fibers on the ECM that had been stiffened by pressure- and cytokine-induced deposition of fibronectin, collagens, proteoglycans, and other ECM components (*Li et al., 2022b*). The conserved TM TRPV4 expression (*Ryskamp et al., 2016*) and TM physiology in mice vs. humans (*Boussommier-Calleja et al., 2012*; *Overby et al., 2014*) suggest that the findings reported might be relevant in a clinical context. The absence of structural and functional visual phenotypes in *Trpv4$^{-/-}$* mice (*Redmon et al., 2024*; *Yarishkin et al., 2018b*; *Jo et al., 2016*) additionally predicts that IOP lowering, suppression of fibrosis, and protection of retinal neurons from pressure by small-molecule TRPV4 antagonists can be implemented without compromising homeostatic IOP regulation (*Sharif, 2023*).

## Methods

### Animals

C57BL/6J mice were from JAX laboratories (Bar Harbor, ME), *Trpv4$^{-/-}$* (*Trpv4$^{tm1.1Ldtk}$ Tg(KRT14-cre/ERT)20Efu/0*; MGI:5544606) mice were a gift from Wolfgang Liedtke (Duke University) (*Liedtke and Friedman, 2003*; *Ryskamp et al., 2011*). The animals were maintained in a pathogen-free facility with a 12-hr light/dark cycle and ad libitum access to food and water, at a temperature of ~22–23°C. Mice were 2–6 months in age prior to LV injection; data from both male and female sexed animals were included in this study.

### Human TM culture

De-identified postmortem eyes from donors with no history of glaucoma (pTM cells) were procured from Utah Lions Eye Bank with written informed consent of the donor's families. TM cells were isolated

**Table 1.** Donor information for primary human trabecular meshwork (pTM) strains used in this study.

| Location | Donor age | Donor sex | Experiments used |
| --- | --- | --- | --- |
| Utah | 55 | M | PCR, Electrophysiology |
| Utah | 76 (a) | F | PCR, WB, Ca$^{2+}$ Img. |
| Utah | 76 (b) | F | PCR, Ca$^{2+}$ Img. |
| Utah | 78 | M | PCR, Ca$^{2+}$ Img. |
| Utah | 64 (a) | F | PCR, WB, Ca$^{2+}$ Img. |
| Utah | 64 (b) | F | PCR, Ca$^{2+}$ Img. |
| Utah | 70 (a) | F | PCR, WB, Ca$^{2+}$ Img., Electrophysiology |
| Utah | 70 (b) | F | PCR |
| Utah | 53 | M | Ca$^{2+}$ Img. |
| Utah | 26 | M | Ca$^{2+}$ Img., Electrophysiology |
| Utah | 73 | F | Ca$^{2+}$ Img. |
| Utah | 56 | M | Ca$^{2+}$ Img. |
| Utah | 73 | M | Ca$^{2+}$ Img. |
| Utah | 80 | M | WB |
| SUNY | 39 | M | Contractility |
| SUNY | 50 | F | Contractility |
| SUNY | 56 | F | Contractility |

from juxtacanalicular and corneoscleral regions as previously described (*Ryskamp et al., 2016*; *Yarishkin et al., 2021*). pTM cells were cultured in Trabecular Meshwork Cell Medium (TMCM; Sciencell Research Laboratories, Carlsbad, CA) in collagen-I-coated culture flasks and glass coverslips at 37°C in a humidified atmosphere with 5% $CO_2$. Fresh media was supplied every 2–3 days. Serum-free (SF) media was mixed as needed by excluding fetal bovine serum (FBS, Sciencell) from the TMCM. A list of all pTM strains used is available in *Table 1*; all cells were used between passages 2 and 4. Cell lines were chosen based on availability at the time of experiments.

TM cell lines were authenticated in accordance with consensus recommendations (*Keller et al., 2018*) and validated as described (*Ryskamp et al., 2016*; *Yarishkin et al., 2021*) with DEX-induced upregulation of MYOC. There was no evidence of mycoplasma infection (i.e., DNA fragmentation/ TUNEL, apoptosis, or reduced cell growth rate).

For contractility experiments, pTM cells were isolated from healthy donor corneal rims discarded after transplant surgery, as previously described (*Li et al., 2022b*; *Li et al., 2022a*; *Bagué et al., 2022*), and cultured according to established protocols (*Keller et al., 2018*; *Stamer et al., 1995*). Three pTM cell strains isolated from healthy donors and validated with dexamethasone-induced myocilin expression were used for contractility experiments. pTM cells were cultured in low-glucose Dulbecco's modified Eagle's medium (DMEM; Thermo Fisher Scientific, Waltham, MA) containing 10% FBS (R&D Systems, Oakwood, GA) and 1% penicillin/streptomycin/glutamine (PSG; Gibco; Thermo Fisher) and maintained at 37°C in a humidified atmosphere with 5% $CO_2$. Fresh media was supplied every 2–3 days.

The experiments were conducted according to the tenets of the Declaration of Helsinki for the use of human tissue.

## Reagents

The TRPV4 antagonist HC-067047 (HC-06) was purchased from Millipore-Sigma (Burlington, MA) or Cayman Biotech (Ann Arbor, MI) and dissolved in DMSO at 20 mM. The TRPV4 agonist GSK1016790A (GSK101; Cayman Biotech) was dissolved in DMSO at 1 mM. Aliquots were diluted into working concentrations (10–25 nM, GSK101; 5–100 µM, HC-06). Recombinant human TGFβ2 protein (R&D Systems) was reconstituted in sterile 4 mM HCl with 0.1% BSA at 20 µg/ml.

**Table 2.** Sequences, product size, and reference numbers for PCR primers used in this study.

| Gene | Forward | Reverse | Product length (bp) | NCBI reference number |
|---|---|---|---|---|
| GAPDH | CTCCTGTTCGACAGTCAGCC | GACTCCGACCTTCACCTTCC | 89 | NM_002046.5 |
| SMAD2 | GGGTTTTGAAGCCGTCTATCAGC | CCAACCACTGTAGAGGTCCATTC | 149 | NM_005901.6 |
| SMAD3 | CAAGTGGCCGCGTGTAAAAA | AGTCCAGAACAGCCGAGTTG | 181 | NM_005902.4 |
| SMAD7 | CTGCTCCCATCCTGTGTGTT | CCTTGGGTTATGACGGACCA | 120 | NM_005904.3 |
| TGFBR2 | AACCTCTAGGCACCCTCCTC | AACCTCTAGGCACCCTCCTC | 100 | NM_001024847.3 |
| FSP1 | GCTTCTTCTTTCTTGGTTTGATCCT | AAGTCCACCTCGTTGTCCCT | 250 | NM_002961.3 |
| SNAIL1 | GGCTCCTTCGTCCTTCTCCTCTAC | CTGGAGATCCTTGGCCTCAGAGAG | 124 | NM_005985.4 |
| CCN2 | CCCCAGACACTGGTTTGAAG | CCCACTGCTCCTAAAGCCAC | 100 | NM_001901.3 |
| YAP1 | ACAGGGAAGTGACTTTGTACA | GCACTGAATATTGCACCCAC | 183 | NM_001130145. |
| FN1 | CTGAAAGACCAGCAGAGGCA | GTGTAGGGGTCAAAGCACGA | 110 | M10905.1 |
| SMA (ACTA2) | GTCACCCACAATGTCCCCAT | GGAATAGCCACGCTCAGTCA | 123 | NM_001141945.2 |
| MYOC | CCACGTGGAGAATCGACACA | TCCAGTGGCCTAGGCAGTAT | 118 | NM_000261.1 |
| TRPV4 | TCCCATTCTTGCTGACCCAC | AGGGCTGTCTGACCTCGATA | 217 | NM_021625.4 |
| PIEZO1 | GGCCAACTTCCTCACCAAGA | GGGTATTTCTTCTCTGTCTC | 106 | NM_001142864.3 |
| TREK1 | AGGGATTTCTACTTGGCGGC | CAAGCACTGTGGGTTTCGTG | 99 | NM_001017424.3 |
| TRPC1 | TGCGTAGATGTGCTTGGGAG | CGTTCCATTAGTTTCTGACAACCG | 107 | X89066.1 |

## Quantitative real-time PCR

Gene-specific primers were used to detect the expression of target genes, as described (*Phuong et al., 2017*). Total RNA was isolated using the Arcturus PicoPure RNA isolation kit (Thermo Fisher Scientific). cDNA was generated from total RNA using qScript XLT cDNA Supermix (Quanta Bio, Beverly, MA). SYBR Green-based real-time PCR was performed with 2X GREEN Master Mix (Apex Bioresearch Products; Houston, TSX). GAPDH was used as an endogenous control to normalize fluorescence signals. Gene expression relative to GAPDH was measured using the comparative CT method ($2^{-[\Delta CT(gene) - \Delta CT(GAPDH)]}$). All genes were assessed in four to eight individual samples taken from three to seven different pTM strains. The primer sequences, expected product length, and gene accession are provided in *Table 2*.

## Western blot

Three SF- or TGFβ2-treated samples were pelleted and pooled together from three different pTM samples within the same condition. To separate membrane proteins from heavier cellular debris, the pooled cell pellets were homogenized in a hypotonic lysis buffer (20 mM Tris-HCl, 3 mM $MgCl_2$, 10 mM NaCl, 10 mM PMSF, 0.5 mM DTT, 20 mM NaF, 2 mM NaV, 0.5 µg/ml leupeptin) before centrifuging at $300 \times g$ for 5 min (4°C). The resulting supernatant was removed and centrifuged again at >12,500 rpm for 30 min to pellet membrane proteins, which were then resuspended in RIPA Buffer (Santa Cruz). Proteins were separated on a 10% SDS–PAGE gel and transferred to polyvinylidene difluoride membranes (Bio-Rad). Membranes were blocked with 5% skim milk/2% BSA in TBST and incubated at 4°C overnight with a primary antibody against TRPV4 (1:250, Alomone Labs #ACC-034; Jerusalem, Israel) or rabbit antibody against β-tubulin (1:2000, Abcam #EPR1330; Waltham, MA). Appropriate secondary antibodies conjugated to HRP were used to visualize protein expression on an iBright CL750 imaging system (Thermo Fisher Scientific). β-Tubulin expression was used to standardize protein levels between samples.

## Calcium Imaging

Primary human TM cells were seeded onto collagen-I-coated coverslips and cultured in TMCM media (ScienCell) as described (*Yarishkin et al., 2021*; *Lakk and Križaj, 2021*). The cells were serum starved for 24 hr followed by SF TMCM with or without TGFβ2 (1 or 5 ng/ml) for 24 hr or 5 days. The cells were loaded with 10 µM of the ratiometric indicator Fura-2 AM $K_d$ at RT = 225 nM (Invitrogen/Thermo Fisher) for 30–60 min. Coverslips were placed in an RC-26G chamber platform (Warner Instrument Corp, Hamden, CT) and perfused with external saline (pH 7.4) (in mM): 80 NaCl, 4.7 KCl, 1.2 $MgCl_2$, 10 D-glucose, 19.1 HEPES sodium salt, 2 $CaCl_2$, and osmolality adjusted to 300 mOsm using D-mannitol. External solutions were delivered via a manually controlled gravity-fed eight-line manifold system, with perfusion speed kept constant to minimize changes in shear. Epifluorescence imaging was performed using an inverted Nikon Ti microscope with a 40× 1.3 N.A. oil objective and Nikon Elements AR software. 340 and 380 nm excitation were delivered by a high-intensity 150 W Xenon arc lamp (Lambda DG-4; Sutter Instruments), high pass-filtered at 510 nm and detected with a 12-bit Delta Evolve camera (Teledyne Photometrics; Tucson, CA). GSK101 10 nM evoked $\Delta[Ca^{2+}]_i$ was assessed as $\Delta R/R$ (dividing the difference between the peak GSK-evoked $F_{340}/F_{380}$ signal during stimulation and baseline $F_{340}/F_{380}$ signal by the baseline $F_{340}/F_{380}$ signal). Every data point represents a separate experimental day and pTM cell strain, each with three to five control and three to five TGFβ2-treated slides tested on the same day. TGFβ2 data points represent the average GSK101 evoked $\Delta R/R$ across all TGFβ2 cells as a % of the average $\Delta R/R$ of control cells from the same cell strain on the same day.

## Collagen hydrogel contraction assay

Rat tail collagen type I (Thermo Fisher Scientific) was prepared at a concentration of 1.5 mg/ml according to the manufacturer's instructions. Five hundred microliters of the hydrogel solution were pipetted into 24-well culture plates. Upon complete collagen polymerization, pTM cells were seeded at $1.5 \times 10^5$ cells/well atop the hydrogels and cultured in DMEM + 10% FBS + 1% PSG for 48 hr to facilitate even cell spreading. Next, constructs were cultured in SF DMEM + 1% PSG supplemented with: (i) control (vehicle: 0.008 mM HCl + 0.0004% BSA; 0.025% DMSO), (ii) TGFβ2 (5 ng/ml; R&D Systems), or (iii and iv) TGFβ2 + HC067047 (5 µM in DMSO) for 36 hr before carefully releasing the hydrogels from the walls using a sterile 10 µl pipette tip to facilitate contraction. The next morning,

fresh SF DMEM + 1% PSG supplemented with 0.0025% DMSO = vehicle was added to groups i–iii; group iv received SF DMEM + 1% PSG supplemented with GSK1016790A (25 nM in DMSO). Plates were longitudinally imaged at 600 dpi resolution with a CanoScan LiDE 300 flatbed scanner at 0, 15, 30, 60, and 120 min. Hydrogel construct size was quantified using FIJI software (National Institutes of Health) (*Schindelin et al., 2012*).

## Electrophysiology

Borosilicate patch-clamp pipettes (WPI) were pulled using a P-2000 horizontal micropipette puller (Sutter Instruments), with a resistance of 6–8 MΩ. The internal solution contained (mM): 125 K-gluconate, 10 KCl, 1.5 $MgCl_2$, 10 HEPES, 10 EGTA, pH 7.4. Patch clamp data were acquired with a Multiclamp 700B amplifier, pClamp 10.6 software, and Digidata 1440A interface (Molecular Devices; San Jose CA), sampled at 5 kHz and analyzed with Clampfit 10.7. Current–voltage relationships were assessed using $V_m$ steps from –100 to +100 mV against a holding potential of –30 mV. Current density was measured as the average current during GSK101 exposure subtracted by the average current from the same cell during baseline perfusion.

## IOP measurements

A TonoLab rebound tonometer (Colonial Medical Supply; Franconia NH) was used to measure IOP of awake mice between 12 and 2 PM. IOP was determined from the mean of 10–20 tonometer readings. Nocturnal measurements were conducted between 9 and 10 PM in awake animals or under 2.5% isoflurane delivered by a Somnosuite isoflurane vaporizer (Kent Scientific; Torrington, CT). After animals recovered from intracameral HC-06/PBS injections, IOP was measured daily. IOP was measured every day for 4–5 consecutive days to confirm a stable return to baseline. IOP data for individual cohorts were binned into weeks of experimental time to group values for analysis.

## Lentiviral transduction

Lentiviral stock for TGFβ2 (C226,228S) was purchased from VectorBuilder Inc (Chicago, IL) (VB170816-1094fnw, pLV[Exp]-CMV> {hTGFB2[NM_003238.3](C226,228S)}) (*Patil et al., 2022*). Scrambled control lentivirus was purchased from SignaGen Laboratories (Frederick, MD) (LM-CMV-Null-Puro). Mice were anesthetized with an intraperitoneal injection of ketamine/xylazine (90 mg/10 mg/kg body weight), followed by eyedrops containing 0.5% proparacaine hydrochloride and 1% tropicamide ophthalmic solution to numb the eyes and dilate the pupils. Anesthetized mice were secured to allow stereotaxic injection of lentivirus. Intravitreal injections were conducted by creating a guide hole with a 30-gauge needle 1–2 mm equatorial of the cornea–scleral border, followed by insertion of a 12° beveled 33-gauge Hamilton syringe, secured to a stereotaxic rig (World Precision Instruments; Sarasota, FL) used to insert the needle 2–3 mm into the eye. Each eye was injected with a 2-µl bolus of lentivirus diluted to $1 \times 10^6$ TU/µl over the course of 1 min, before the needle was quickly drawn and the pilot hole treated with erythromycin ophthalmic ointment USP (Bausch & Lomb; Laval, Canada). The efficiency of LV-TGFβ2 OHT induction in WT animals was close to 100%. No differences in observable health post-injection were detected between wild-type and $Trpv4^{-/-}$ animals or LV-Ctrl and LV-TGFβ2 injected animals.

## Intracameral microinjections

Mice were anesthetized and treated with eyedrops as above, before being placed on an isothermal heating pad. HC-06 (100 µM) or PBS with DMSO (0.5%) as a vehicle were injected into the anterior chamber using a blunt tip Hamilton syringe through a guide hole made using a 30-gauge needle. At the end of each injection, a small air bubble was introduced to seal the cornea and minimize fluid outflow. 0.5% erythromycin ophthalmic ointment USP (Bausch & Lomb) was applied to the eye after the procedure. Intracameral injections were not associated with observable inflammation, corneal opacity, or behavioral changes. For the nocturnal IOP experiments in *Figure 6*, both eyes of two animals were injected with PBS while two were injected with HC-06. When OHT was stably reestablished a week post-injection, the treatment groups were switched, and experiments repeated, resulting in four eyes/treatment group for *Figure 6E, F*.

## Statistical analysis

GraphPad Prism 9 was used for statistical analysis. Means are plotted ± SEM unless otherwise noted. One-sample *t*-tests were used to determine whether TGFβ2-treated groups were significantly

different than untreated control groups, while one- or two-way ANOVA along with Tukey or Bonferroni's multiple comparisons test were used to compare multiple groups.

## Study approval

The animal experimental protocols were conducted in accordance with the NIH Guide for the Care and Use of Laboratory Animals and the ARVO Statement for the Use of Animals in Ophthalmic and Vision Research and were approved by the Institutional Animal Care and Use Committee at the University of Utah (protocol 25-00001905).

## Acknowledgements

We thank Dr. Paloma Liton (Duke University) for the generous gift of LV-TGFβ2$^{C226,228S}$ lentivirus stock used for the pilot stages of this experiment, and Dr. Gulab Zode for the availability of the LV-TGFβ2$^{C226,228S}$ construct on Vectorbuilder. We additionally thank Dr. Wolfgang Liedtke (Duke University and Regeneron) for *Trpv4$^{-/-}$* mice. The study was supported by the National Institutes of Health (T32EY024234 to CNR and DK, R01EY022076, R0EY1031817, P30EY014800 to DK, R01EY034096, R21EY036189 to SH, R01EY022359, R01EY005722 to WDS), Crandall Glaucoma Initiative, Stauss-Rankin Foundation, and Unrestricted Grants from Research to Prevent Blindness to Ophthalmology Departments Duke University, SUNY Upstate Medical University, and the University of Utah. Schematics were made with Biorender.com.

## Additional information

### Competing interests

David Krizaj: Co-founder of TMClear and co-inventor of patents (US 2015/0133411, US20230026696) related to the development of cornea permeant TRPV4 channel antagonists; the patents were licensed to TMClear by the University of Utah. The other authors declare that no competing interests exist.

### Funding

| Funder | Grant reference number | Author |
| --- | --- | --- |
| National Eye Institute | T32EY024234 | Christopher Nass Rudzitis David Krizaj |
| National Eye Institute | R01EY022076 | David Krizaj |
| National Eye Institute | R0EY1031817 | David Krizaj |
| National Eye Institute | P30EY014800 | David Krizaj |
| National Eye Institute | R01EY034096 | Samuel Herberg |
| National Eye Institute | R01EY022359 | W Daniel Stamer |
| National Eye Institute | R01EY005722 | W Daniel Stamer |
| Crandall Glaucoma Initiative | | David Krizaj |
| Stauss-Ranking Foundation | | David Krizaj |
| Research to Prevent Blindness | | W Daniel Stamer David Krizaj Samuel Herberg |
| National Institutes of Health | R21EY036189 | Samuel Herberg |

The funders had no role in study design, data collection, and interpretation, or the decision to submit the work for publication.

## Author contributions
Christopher Nass Rudzitis, Conceptualization, Data curation, Software, Formal analysis, Funding acquisition, Validation, Investigation, Visualization, Methodology, Writing – original draft; Monika Lakk, Sarah N Redmon, Formal analysis, Investigation, Methodology; Ayushi Singh, Yun-Ting Tseng, Formal analysis, Investigation; Denisa Kirdajová, Investigation; Michael L De Ieso, Conceptualization, Methodology; W Daniel Stamer, Conceptualization; Samuel Herberg, Conceptualization, Data curation, Formal analysis, Investigation, Methodology; David Krizaj, Conceptualization, Resources, Data curation, Formal analysis, Supervision, Funding acquisition, Investigation, Methodology, Writing – original draft, Project administration, Writing – review and editing

## Author ORCIDs
David Krizaj (ID) https://orcid.org/0000-0003-4468-3029

## Ethics
Animal handling, anesthetic procedures, and experiments were performed in accordance with the NIH Guide for the Care and Use of Laboratory Animals and the ARVO Statement for the Use of Animals in Ophthalmic and Vision Research and were approved by the Institutional Animal Care and Use Committees at the University of Utah (protocol 25-00001905).

Reviewer #1 (public review): https://doi.org/10.7554/eLife.104894.3.sa1
Reviewer #2 (public review): https://doi.org/10.7554/eLife.104894.3.sa2
Author response https://doi.org/10.7554/eLife.104894.3.sa3

---

# Additional files

## Supplementary files
MDAR checklist

## Data availability
Individual datapoints for in vivo figures, and unedited/uncropped annotated western blot images are included in the supplementary data files for this manuscript. Source data has been provided for Figures 5 and 6. Further information about the data presented in this manuscript is available from the corresponding authors upon reasonable request.

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
