## [Editor Report · eLife Assessment]

This **fundamental** work extends our understanding of the role of TGFβ2 as a modulator of mechanosensing in the eye and identifies the TRPV4 ion channel as a common regulator of Trabecular Meshwork (TM) contractility and pathological OHT and the data and evidence provided are **convincing**. This work will clearly be of interest to researchers investigating the role of mechanosensors in the TM and may underpin future research into treatments that aim to lower intra ocular pressure. This work will additionally be of interest to the growing field of researchers investigating the regulation of force sensing via ion channels and their roles in health and disease, in particular the ion channel TRPV4.

---

## [Referee Report · Reviewer #1 (public review)]

Summary:

This comprehensive study employed molecular, optical, electrophysiological and tonometric strategies to establish the role of TGFβ2 in transcription and functional expression of mechanosensitive channel isoforms alongside studies of TM contractility in biomimetic hydrogels, and intraocular pressure regulation in a mouse model of TGFβ2 -induced ocular hypertension. TGFβ2 upregulated expression of TRPV4 and PIEZO1 transcripts and time-dependently augmented functional TRPV4 activation. TRPV4 activation induced TM contractility whereas pharmacological inhibition suppressed TGFβ2-induced hypercontractility and abrogated ocular hypertension in eyes overexpressing TGFβ2. Trpv4-/- mice resisted TGFβ2-driven increases in IOP. These data establish a fundamental role of TGFβ as a modulator of mechanosensing and identifies TRPV4 channel as a common mechanism for TM contractility and pathological ocular hypertension.

The manuscript is very well written and details the important function of TRPV4 in TM cell function. These data provide novel therapeutic targets and potential for disease-altering therapeutics.

---

## [Referee Report · Reviewer #2 (public review)]

The manuscript by Christopher N. Rudzitis et al. describes the role of TGFβ2 in the transcription and functional expression of mechanosensitive channel isoforms, alongside studies on TM contractility in biomimetic hydrogels and intraocular pressure. Overall, it is a very interesting study, nicely designed, and will contribute to the available literature on TRPV4 sensitivity to mechanical forces.

---

## [Author Response]

The following is the authors’ response to the original reviews

**Reviewer #1 (Public review):**
The experimental rigor and design of the noctural IOP experiments was weak with low n values and differing methods of IOP measurement (conscious versus anesthetized). The same method of IOP measurement needs to be used for all measurements to make any conclusions on the circadian patterns of IOP in each condition.

One of the goals of our study was to confirm the results from the Patel et al (2021; PMID33853948) study, which in which nocturnal IOP measurements were conducted in anesthetized mice and diurnal IOP measurements in awake animals but we agree with both Reviewers that IOP should be measured under identical experimental conditions. Parenthetically, the number of animals per each treatment paradigm in the original version (N = 4) was sufficient to produce statistical significance for diurnal control vs diurnal TGFB, and diurnal control vs nocturnal control conditions.

To address the comment, we generated an additional cohort of TGFb2-expressing mice (N = 6) in which nocturnal and diurnal measurements were performed in awake animals. The results are shown in the revised Figure 6. Similar to the anesthetized cohort, the diurnal IOP in Lv-TGFB2 mice was statistically indistinguishable from the nocturnal value, indicating that TGFB2-induced OHT is not additive to physiological (circadian) OHT. The TRPV4-dependence of ocular hypertension induced by physiological and pathological methods suggests that the channel functions as a final common mechanism for ocular hypertension.

**Reviewer #2 (Public review):**
Figure 1A-C. Often there is a difference between the massage (message?, op. authors) and transcript data. I recommend the authors to confirm with qPCR data with another mode of protein measurements.

We are not sure we understand the Reviewer’s comment regarding the “difference between the message and transcript data” but note that the mRNA data shown in panels A & B are confirmatory of previously published transcriptomic and proteomic screens (eg, Fleenor et al., IOVS 2006; Bollinger et al., IOVS 2011; Callaghan et al., Scientific Reports 2022; Li et al., Current Eye Research 2022 etc) and were included to show that the transcriptional response of canonical SMAD and pro-fibrotic genes unfolds as predicted from previous work. With regard to TRPV4 signaling, we expand transcriptomic data with protein analysis (Western blots) and functional analyses (measurements of TRPV4-mediated current and calcium imaging). Transcriptomic, protein expression, electrophysiological and imaging experiments revealed a remarkable consistency in TGFB2-dependence of gene (Fig. 1C) and protein expression (Fig. 1D), transmembrane current (Fig. 3C) and intracellular calcium (Fig. 2).

Parenthetically, we attempted to get a sense for the TGFB2-dependence of Piezo1 protein expression by conducting Western blots with multiple antibodies and experimental conditions. These efforts were unsuccessful, presumably due to the complexity (30-40 TM domains) and large molecular weight (280-300 kDa) of the protein. We note, however, that Piezo1 signaling cannot account for the observed OHT given that studies by us and others (Yarishkin et al., 2021, PMID: 33226641 and Zhu et al., 2021; PMID: 33532718) associated Piezo1 signaling with facility increases. The revised m/s reads: “The suppression of outflow facility by Piezo1 inhibitors applied under in vitro and in vivo conditions (39, 81) instead suggests that Piezo1 opposes the hypertensive functions of TRPV4.” The preprint by Redmon et al. (bioRxiv 2024, PMID 39041037) expands the TRPV4-dependence of OHT to microbead-induced, steroid-induced and nocturnal models of OHT to indicate that TRPV4 functions as a universal driver of elevated IOP. We reiterate this in the revised Discussion.

Does direct TRPV4 activation also induce the expression of these markers? Does inhibition of TRPV4, after TGF-β treatment, prevent the expression of these markers? Is TRPV4 acting downstream of this response?

A RNASeq study conducted by us (Rudzitis et al., under review) suggests that the agonist GSK101 has minimal effect on the fibrotic and canonical pathways shown in panels A and B. These data are beyond the scope of the present study. They will be published elsewhere, however, we include the data associated with genes depicted in panels A and B for the reviewer at the end of this Response.

We conducted an additional series of experiments to test whether TGFB2-induced upregulation of the TRPV4 and Piezo1 genes is itself TRPV4-dependent. As shown in the new SFig. 1, upregulation of the two genes is unaffected by TRPV4 inhibition.

Figure 1D. Beta tubulin is not a membrane marker. Having staining of b tubulin in membrane fraction shows contamination from the cytoplasm. Does the overall expression also increase?

b-tubulin associates with the plasma membrane by binding to integral membrane proteins in the plasma and organellar membranes through palmitoylation and attachment to linker proteins and as an integral component of exocytotic vesicles (Wolff, BBA 2009; Hogerheide et al., PNAS 2017). The protein is often used as a loading control for the TRPV4 protein (please see https://www.cellsignal.com/products/primary-antibodies/trpv4-antibody/65893; Grove et al., Science Signaling 2019 and Moore et al., PNAS 2013). Parenthetically, our RNASeq studies did not find modulation of b-tubulin expression by TGFβ2 [CNR and DK, unpublished observations].

We examined the overall (cytosolic and membrane) TRPV4 expression and observed, similarly to the membrane fraction alone (Figure 2), upregulation following cytokine stimulation:

**Author response image 1. sa3fig1:** Western blot, total protection extract from control and TGFb2-treated TM cells [Alomone antibody].

These results in our estimation do not add to the overall narrative and were not included into the paper.

Figure 4A: it is not very clear. I recommend including a zoom image or better resolution image.

We include a whole-page image as the new SFigure 4.

Figure 5B and 6B. Why there is a difference between groups in pre-injection panel. As Figure 5A, in pre-injection, there is no difference between LV-TGFβ and LV-control while in 5B there is a significant difference between these groups.

We revised Figure Legends to clarify that “pre-injection” in Figures 5B and 6B refers to IOP measurements before the intracameral injection of HC-06 not pre-injection of lentiviral constructs.

Discussion section. Line 279: "TRPV4 channels in cells treated with TGFβ2 are likely to be constitutively active" ... needs to be discussed further.

We rewrote the paragraph to clarify that TRPV4 is a thermosensitive channel that is expected to be constitutively active at the incubator temperature:

“The effectiveness of TRPV4 inhibition in suppressing TGFB2-induced contractility (Fig. 4) is consistent with constitutive activation of TRPV4 channels in incubator-cultured cells. TRPV4 is a thermosensitive channel (Q10 ~10). Mouse TRPV4 is activated by physiological temperatures (Chung et al., 2003; Shibasaki et al., 2007) with peak activation between ~34 - 37oC (Guler et al., 2003). The several-fold increase in functional expression of the channel in TGFB2-treated cells (Fig. 2) would be expected to promote tonic influx of Ca2+ and Ca2+-dependent cellular signaling. The abrogation of the contractile response in the presence of HC-06 indicates that TRPV4-mediated Ca2+ influx represents the principal source of calcium that drives the contractile response. Consistent with this, supplementation with the agonist GSK101 was sufficient to evoke TM contraction (Fig. 4B).”

Line 280: "The residual contractility in HC-06-treated cells may reflect TGFβ2-mediated contributions from Piezo1." Piezo1 has a low threshold for mechanosensitivity. How do the authors discuss the observation that, in the presence of Piezo1, TRPV4 has a more prominent mechanosensory function? Is this tied to TGFβ signalling?

This is an interesting question. Our macroscopic and single channel recordings of Piezo1 activity in TM cells recapitulate the time course published in the original Coste et al. (2010) study, showing the channel inactivates within 10-100 msec (Yarishkin et al., 2021). Thus, it is likely that the channel is largely inactivated during chronic ocular hypertension. Indeed, it has been suggested that resting membrane tension alone may be sufficient to inactivate Piezo1 (Lewis and Grandl, 2015), with cells grown on stiff substrates (e.g., under our experimental conditions) experiencing almost complete Piezo1 inactivation. We propose that the primary function of Piezo channels may be to sense and transduce transient mechanical loading. The remarkable IOP-lowering effectiveness of TRPV4 antagonists and knockdown indicates that - in contrast to Piezo1 - TRPV4 activation is sustained.

**Recommendations for the authors:**

**Reviewer #1 (Recommendations for the authors):**
The complete strain name for the Trpv4-/- mice are missing.

Corrected.

The layout for Figure 6 is confusing as HC-06 was only used in panels B and C but the labels are above panel A.

Corrected.

**Reviewer #2 (Recommendations for the authors):**
Only two mice were used for the noctural IOP experiments. Justification for retreating the same mice in opposite eyes and counting it as n=4 is not rigorous or justified.

The number of mice investigated in the original submission was four. In Week 1, two mice underwent PBS injections and 2 two mice were treated with HC-06. After the baseline was re-established in Week 2, the treatments were reversed.

We supplemented these numbers with an additional cohort of 6 mice, with identical results re: nocturnal vs diurnal IOP. These data are presented in the revised Figure 6.

Why are daytime IOPs measured in awake mice but noctural IOP's measured in isoflurane anesthetized mice? Anesthesia is well known to effect IOP and using two different methods could alter the results, especially when comparing between the groups. This could be why you did not see a noctural rise in the TGFB injected eyes. The same method needs to be used for all measurements to make any conclusions on the circadian patterns of IOP in each condition.

This is a good point, please see our response above.